# MASS: MULTI-AGENT SIMULATION SCALING FOR PORTFOLIO CONSTRUCTION

## ABSTRACT

The application of LLM-based agents in financial investment has shown significant promise, yet existing approaches often require intermediate steps like predicting individual stock movements or rely on predefined, static workflows. These limitations restrict their adaptability and effectiveness in constructing optimal portfolios. In this paper, we introduce the Multi-Agent Scaling Simulation (MASS), a novel framework that leverages multi-agent simulation for direct, end-to-end portfolio construction. At its core, MASS employs a backward optimization process to dynamically learn the optimal distribution of heterogeneous agents, enabling the system to adapt to evolving market regimes. A key finding enabled by our framework is the exploration of the scaling effect for portfolio construction: we demonstrate that as the number of agents increases exponentially (up to 512), the aggregated decisions yield progressively higher excess returns. Extensive experiments are conducted on a challenging, proprietary cross-market dataset from 2023, showing that MASS gains enhanced performance over nine state-of-the-art baselines. Further backtesting, stability analyses, the experiment on data leakage concerns, and case studies validate its enhanced profitability and robustness. We have open-sourced our code, dataset, and training snapshots at `https://anonymous.4open.science/r/MASS-AC96`[1] to foster further research.

## 1 INTRODUCTION

The application of LLM-based agents in investment analysis has recently garnered significant attention from both academia and industry (Tang et al., 2025; Xiao et al., 2025b; Li et al., 2025b). By assigning LLMs with distinct roles and providing them with relevant financial context, researchers have developed agent-based systems to tackle complex tasks such as alpha factor mining (Cao et al., 2025; Li et al., 2025b) and stock trend prediction (Koa et al., 2024; Yu et al., 2024). These pioneering efforts highlight the potential of LLMs to process and reason over vast amounts of multi-modal financial data, including news, reports, and market indicators.

Despite their promise, existing LLM-based approaches in finance exhibit two primary limitations. First, many systems are designed for individual stock forecasting (Koa et al., 2024; Yu et al., 2024; Xiao et al., 2025b). While useful, predicting the movement of single stocks does not directly translate to constructing an optimal portfolio, which requires a holistic assessment of asset correlations, market sentiment, and risk diversification. Second, these systems typically rely on predefined procedural workflows to orchestrate agent interactions (Guo et al., 2024). This reliance on static, pre-programmed processes limits their ability to adapt to the highly dynamic and non-stationary nature of financial markets, potentially compromising their performance during market regime shifts.

In this paper, we introduce the Multi-Agent Scaling Simulation (**MASS**) to address these challenges. MASS shifts the paradigm from individual stock prediction to direct portfolio construction by simulating a market of heterogeneous investor agents. Instead of relying on static workflows, MASS introduces a backward optimization process. This mechanism uses historical market data to dynamically learn the optimal distribution of agent types that maximizes portfolio returns, allowing the system to adapt its strategy. This approach provides MASS with three key advantages: (1)

---

[1]We also provide a fully anonymized GitHub mirror as a backup: `https://github.com/anonymous3728/MASS_anonymous`.

It leverages aggregated information from a multi-agent simulation for direct, end-to-end portfolio construction, bypassing intermediate steps like individual stock prediction; (2) It replaces predefined workflows with a data-driven optimization process, enhancing adaptability and performance; and (3) It enables us to explore the multi-agent scaling effect for portfolio construction: as the number of agents increases exponentially, the system's decisions achieve higher excess returns. To the best of our knowledge, MASS is the first work to scale multi-agent simulation for this task up to 512 agents.

To rigorously evaluate MASS, we construct a challenging dataset from the Chinese A-share market for 2023, a period characterized by high volatility and two major market regime shifts. On this primary dataset, spanning the *SSE50*, *CSI 300*, and *ChiNext 100* indices, extensive experiments show that MASS consistently outperforms nine state-of-the-art baselines. Backtesting simulations further confirm its ability to generate higher excess returns with lower drawdowns. To address concerns of data leakage, we validate our findings on unseen data from Q1 2025, a period after the knowledge cutoff of our LLM. MASS maintains its effectiveness on this future data, and demonstrates robustness when implemented with different LLM backbones. Finally, a series of in-depth analyses validate our core design: we demonstrate a multi-agent scaling effect, where performance improves as the number of agents scales up to 512; ablation studies underscore the criticality of our backward optimization mechanism; and visualizations of the agent distribution reveal the model's adaptive response to market shifts. MASS's generalizability is also briefly verified on the US stock market.

In summary, this paper makes the following contributions:

- We introduce MASS, leveraging multi-agent simulation with end-to-end backward optimization for decision-making in portfolio construction.

- To our best knowledge, we are the first to explore and demonstrate a scaling effect in multi-agent simulation for portfolio construction, expanding the number of agents up to 512.

- Extensive experiments show that MASS gains enhanced performance, delivering consistent excess returns, scalability, and stability. We also address potential data leakage concerns and validate our simulation's effectiveness through visualization.

- We have introduced and released a comprehensive, realistic, and rich dataset, along with our code and training snapshots, to facilitate future research in this domain.

## 2 RELATED WORK

This section reviews related work across three key areas to contextualize our research. We first discuss our primary research domain: existing investment analysis approaches within the financial market. We then survey the landscape of LLM-based multi-agent systems, which constitute our methodological approach. Finally, we cover the emerging research on scaling effects for multi-agent systems, a significant finding that informs our understanding of system performance.

### 2.1 INVESTMENT ANALYSIS

Investment analysis research traditionally focuses on two main tracks: formulaic alpha mining and stock price trend prediction. Alpha mining aims to discover mathematical expressions from financial data that predict future returns, using techniques like genetic algorithms (Chen et al., 2021), deep reinforcement learning (Yu et al., 2023; Shi et al., 2025a), and more recently, LLM-based agents (Tang et al., 2025; Cao et al., 2025; Ding et al., 2025; Li et al., 2025b; Shi et al., 2025b). Stock price trend prediction employs methods ranging from traditional time-series analysis (Choi, 2018), factor-model learning Duan et al. (2022); Wei et al. (2023), deep learning models (Yoo et al., 2021; Xu et al., 2021; Luo et al., 2023; Du et al., 2024a; Li et al., 2024a; Yang et al., 2025a; Chen et al., 2025). reinforcement learning models (Niu et al., 2022; Yuan et al., 2025) to the latest LLM-based agents (Koa et al., 2024; Xiao et al., 2025b; Zhang et al., 2024c) and foundation model training (Liu et al., 2025; Xiao et al., 2025a; Shi et al., 2025c). While effective to a degree, alpha mining often treats the market monolithically, overlooking stock-specific idiosyncrasies, and factor-model learning might struggle to utilize rich data sources with various modalities, while most trend prediction methods focus on individual assets rather than portfolio-level optimization. Furthermore, many recent LLM-agent approaches rely on fixed, predefined workflows, limiting their adaptability. Additionally,

LLMs trained on massive historical data introduce the risk of data leakage, as the historical data may encapsulate past market information.

MASS distinguishes itself from these works by shifting the focus from individual stock prediction or factor mining to the direct task of portfolio construction. Unlike methods that rely on predefined workflows, MASS employs a data-driven, end-to-end optimization framework to dynamically infer the underlying distribution of investor archetypes that leads to optimal portfolio performance. This simulation-based approach allows MASS to holistically model market dynamics and adapt to changing conditions, offering superior performance and market adaptability compared to forecasting individual asset movements in isolation.

## 2.2 LLM-BASED MULTI-AGENT SYSTEMS

LLM-based multi-agent systems (MAS) are broadly classified into two categories: *Simulation* and *Application* (Guo et al., 2024). Simulation-focused MAS are used to model emergent social (Park et al., 2023), economic (Zhao et al., 2024; Li et al., 2023b), or psychological phenomena (Kovac et al., 2023; Zhang et al., 2024b). Their primary goal is to validate existing theories or generate analytical insights. In contrast, Application-focused MAS employ specialized agents organized in structures like layers (Liu et al., 2024) or centralized hierarchies (Qian et al., 2025) to collaboratively execute specific tasks, such as software development (Li et al., 2023a) or scientific debate (Du et al., 2024b). These systems typically follow predefined procedural workflows to ensure efficient coordination.

MASS bridges the gap between these two categories. Compared to existing simulations, which are primarily used for analysis, MASS utilizes the aggregated output of its simulation for concrete, real-world decision-making, thereby expanding the practical boundaries of multi-agent simulation. Unlike existing applications that depend on rigid, predefined processes, MASS leverages a data-driven, end-to-end backward optimization mechanism. This allows the system to learn its own optimal collaborative strategy from market feedback, resulting in superior performance and adaptability without the need for hand-crafted workflows.

## 2.3 SCALING EFFECTS IN MULTI-AGENT SYSTEMS

The study of scaling effects—predictable performance improvements with increased model size, data, or compute—is a key component of modern LLM research (Kaplan et al., 2020). One notable study explores cooperative scaling effects for various predefined agent architectures (e.g., linear, tree), expanding the agent count up to 64 (Qian et al., 2025). Another recent work (Dang et al., 2025) proposes an evolving orchestration where an RL-trained puppeteer dynamically organizes agents into cost-effective collaboration topologies, enhancing scaling in MAS.

As for scaling effects in multi-agent systems (MAS) for financial simulation, several recent works provide important context. While Mars (Li et al., 2025a) is a relevant work in market simulation, our approaches differ in scope. Mars provides a micro-level simulation by modeling the order book and the dynamics of individual transactions. In contrast, our work takes a macro-level perspective, simulating the collective behavior of investors as market participants. Other notable works like StockAgent (Zhang et al., 2024a) and TwinMarket (Yang et al., 2025b) also use LLMs to simulate investor behavior, with StockAgent focusing on responses to external factors and TwinMarket leveraging dynamic social networks. However, these frameworks primarily focus on simulating emergent market phenomena for analytical insight. The key differentiator of MASS is its design as an *application-oriented framework* centered around a backward optimization process. Instead of only observing emergent behavior, MASS uses the simulation's aggregated output to make concrete investment decisions and then leverages real market feedback to learn the optimal way to combine agent opinions. This approach effectively shifts the paradigm from analyzing a simulation to actively optimizing a simulation for a real-world investment task.

In contrast, MASS introduces and investigates a scaling paradigm for multi-agent decision-making. Our scaling effect does not rely on a prescribed form of cooperation. Instead, each agent is given a partial view of the market, and as the number of agents increases, the system's collective awareness of the market grows. The core challenge, which we address via our backward optimization process, is learning how to aggregate this distributed intelligence to achieve a specific real-world objective (i.e., maximizing portfolio returns). MASS is the first work to demonstrate this scaling effect in a financial

application, expanding the number of simulated agents to 512 and showing a clear correlation between agent scale and investment performance.

## 3 METHOD

In this section, we introduce MASS, a novel framework that formulates portfolio construction as a dynamic online learning problem. The core idea is to simulate a market of heterogeneous investor agents and learn to optimally aggregate their diverse decisions. MASS operates in a daily cycle of two key processes: **Forward Propagation**, where agents generate investment signals for the current day, and **Backward Optimization**, which refines the model by learning from the previous day's outcomes. This adaptive loop, illustrated in Figure 1, allows MASS to continuously adjust to evolving market conditions. The overall procedure is formalized in Algorithm A.1.

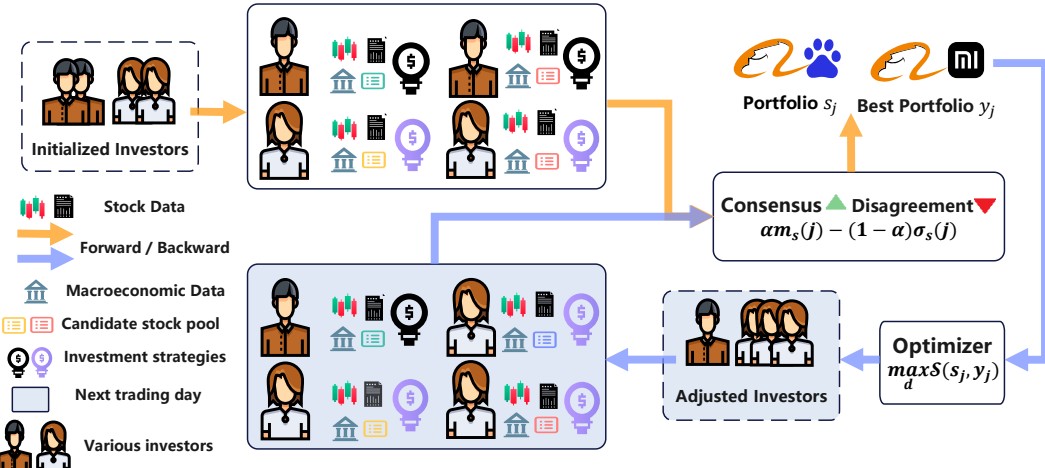

Figure 1: MASS operates in a loop consisting of forward propagation and backward optimization. In the forward propagation, MASS initializes investors using the previous day's investor distribution along with today's stock and macroeconomic data. It then constructs portfolios $s_j$ based on the market disagreement hypothesis. During backward optimization, an optimizer updates the investor distribution, which is then passed to the next trading day.

### 3.1 FORWARD PROPAGATION

The forward propagation process simulates market activity on a given day $j$ to produce a signal that guides the construction of the portfolio. This involves initializing a diverse population of agents, executing their investment strategies, and aggregating their collective decisions based on a multi-modal stock dataset $\mathcal{X}$.

#### 3.1.1 INVESTOR INITIALIZATION

To capture the diverse perspectives within a real market, MASS initializes a population of $N = n^{\text{type}} \times n^{\text{inv}}$ agents. These agents are categorized into $n^{\text{type}}$ distinct types, each embodying a unique investment style (e.g., style outline, risk appetite, rationality). This heterogeneity is crucial for creating a rich and realistic simulation. Each agent type $i$ is provided access to a specific subset of multi-modal data $\mathcal{X}_i \subset \mathcal{X}$. Furthermore, to model the practical constraint that no single investor can monitor the entire market, each individual agent $(i, k)$ is assigned a static, random subset of stocks, denoted as $\text{Pool}(i, k)$, where $|\text{Pool}(i, k)| = n^{\text{sel}}$. This design choice also manages the context length limitations of the underlying LLM. The design details of investor initialization are in Appendix A.3.2.

### 3.1.2 INVESTMENT STRATEGY EXECUTION

On each trading day $j$, agents first formulate a daily strategy and then make investment decisions. First, to ensure strategies are adaptive to the prevailing economic climate, each agent type $i$ generates a daily investment strategy by interpreting the latest macroeconomic data $\mathbf{M}_j$ within the context of its intrinsic style. This is performed by an LLM-based function $F_1$:

$$\text{Strategy}_{i,j} = F_1\big(\text{StyleDesc}_i, \mathbf{M}_j\big) \tag{1}$$

where $\text{StyleDesc}_i$ is the textual description of agent type $i$'s investment philosophy.

Next, each agent $(i, k)$ applies this daily strategy to its observable stock universe $\text{Pool}(i, k)$. The agent analyzes the relevant features for these stocks and selects a subset for investment. This decision is modeled by a second LLM-based function $F_2$:

$$\text{Codes}_{i,k,j} = F_2\Big(\text{Strategy}_{i,j}, \ \{\text{data for } s \in \text{Pool}(i, k)\}, \ \text{StyleDesc}_i\Big) \tag{2}$$

where $\text{Codes}_{i,k,j} \subseteq \text{Pool}(i, k)$ is the set of stocks selected by agent $(i, k)$ on day $j$. The design details of this section are provided in Appendix A.3.3.

### 3.1.3 SCORE AGGREGATION

To derive an actionable signal for each stock, we aggregate the decisions from all $N$ agents. Our aggregation strategy is grounded in the market disagreement hypothesis (Miller, 1977; Diether et al., 2002), which posits that stocks with high consensus and low disagreement among investors tend to yield higher future returns. This provides a theoretically sound basis for combining agent outputs. We provide more details about market disagreement hypothesis on Appendix A.3.1.

Let $V_{i,s,j}$ be the fraction of agents of type $i$ that selected stock $s$ on day $j$. Let $\mathbf{d}_{j-1} = [d_{1,j-1}, \ldots, d_{n^{\text{type}},j-1}]^\top$ be the distribution of agent types, optimized from the previous day. We quantify consensus and disagreement for each stock $s$ by computing the weighted mean ($m_s$) and weighted standard deviation ($\sigma_s$) of selections across all agent types:

$$m_s(j) = \sum_{i=1}^{n^{\text{type}}} d_{i,j-1} \cdot V_{i,s,j} \quad \text{(Consensus)} \tag{3a}$$

$$\sigma_s(j) = \sqrt{\sum_{i=1}^{n^{\text{type}}} d_{i,j-1}(V_{i,s,j} - m_s(j))^2} \quad \text{(Disagreement)} \tag{3b}$$

The final signal for each stock integrates these two components, rewarding consensus and penalizing disagreement:

$$\text{Signal}(s, j) = \alpha \cdot m_s(j) - (1 - \alpha) \cdot \sigma_s(j) \tag{4}$$

where $\alpha \in [0, 1]$ is a hyperparameter balancing the two effects. This signal is then used to rank stocks and construct the daily portfolio $\mathbf{P}_j$.

### 3.2 BACKWARD OPTIMIZATION

A key innovation of MASS is its ability to adapt to changing market regimes. This is achieved through the backward optimization process, which dynamically adjusts the agent type distribution $\mathbf{d}_j$ at the end of each day $j$. The objective is to find the distribution that would have yielded the best performance over a recent historical window, ensuring the model continuously learns from market feedback.

Specifically, at the end of day $j$, we define a look-back window of size $\omega_{\text{opt}}$. We use the agent decisions $\{V_{i,s,t}\}$ and the actual market returns $\{\mathbf{Y}_t\}$ for the period $t \in [j - \omega_{\text{opt}} + 1, j]$. For any candidate distribution $\mathbf{d}$, we can compute the historical signals $\text{Signal}_{\mathbf{d}}(s, t)$ for this period. The goal is to find the optimal distribution $\mathbf{d}_j$ that maximizes the correlation between these historical signals and the actual returns. This is formulated as an optimization problem:

$$\mathbf{d}_j = \arg\max_{\mathbf{d} \in \Delta^{n^{\text{type}}-1}} \mathcal{S}\left(\{\text{Signal}_{\mathbf{d}}(:,t)\}_{t=j-\omega_{\text{opt}}+1}^{j}, \ \{\mathbf{Y}_t\}_{t=j-\omega_{\text{opt}}+1}^{j}\right) \tag{5}$$

where $\Delta^{n^{\text{type}}-1}$ is the probability simplex, and $\mathcal{S}$ is a similarity metric such as the Rank Information Coefficient (RIC). We employ simulated annealing (Kirkpatrick et al., 1983) as the optimizer $\mathcal{O}$ to solve this problem. The resulting distribution $\mathbf{d}_j$ is then carried forward to the next day's forward propagation step (Eq. 3), completing the online learning cycle.

# 4 EVALUATION

Table 1: Comparisons with baselines and the experiment on data leakage concern. MASS outperforms all others across all 3 stock pools. The best performance in each column is highlighted in **bold**. For more evaluation metrics on portfolio construction and evaluation results on the US stock market, please refer to Appendix A.5. All results are performed in percent.

| Method | SSE50 | | | | CSI 300 | | | | Chi Next 100 | | | |
|---|---|---|---|---|---|---|---|---|---|---|---|---|
| | RIC | RICIR | IC | ICIR | RIC | RICIR | IC | ICIR | RIC | RICIR | IC | ICIR |
| **Main Experiments (Throughout 2023)** | | | | | | | | | | | | |
| Proxy Indicator (Diether et al., 2002) | 3.82 | 19.73 | 2.89 | 16.63 | 3.84 | 30.44 | 3.60 | 27.03 | -0.94 | -7.05 | 0.16 | 1.29 |
| LightGBM (Ke et al., 2017) | 3.25 | 21.78 | 4.51 | 27.30 | 5.20 | 36.06 | 3.19 | 23.62 | 2.94 | 30.69 | 0.88 | 8.70 |
| FactorVAE (Duan et al., 2022) | 5.05 | 38.27 | 4.89 | 26.56 | 4.95 | 34.89 | 4.16 | 31.13 | 3.98 | 28.69 | 4.03 | 29.35 |
| HireVAE (Wei et al., 2023) | 5.17 | 29.06 | 5.02 | 29.93 | 5.23 | 36.21 | 4.22 | 31.08 | 4.03 | 32.25 | 4.14 | 30.08 |
| DTML (Yoo et al., 2021) | 5.04 | 28.15 | 4.93 | 26.71 | 4.91 | 35.72 | 4.17 | 31.10 | 3.45 | 26.55 | 3.21 | 21.97 |
| MASTER (Li et al., 2024b) | 5.13 | 28.37 | 4.97 | 27.01 | 5.01 | 35.47 | 4.23 | 30.78 | 3.92 | 31.03 | 4.07 | 28.62 |
| SEP (Koa et al., 2024) | 4.79 | 27.56 | 4.16 | 26.40 | 3.83 | 5.42 | 0.61 | 7.65 | 4.81 | 34.88 | 5.29 | 36.98 |
| FinCON (Yu et al., 2024) | 4.88 | 26.18 | 4.35 | 25.67 | 0.70 | 9.57 | 0.96 | 13.42 | 5.01 | 37.18 | 5.53 | 40.54 |
| TradingAgents (Xiao et al., 2025b) | 4.92 | 27.71 | 4.33 | 25.69 | 3.01 | 10.14 | 1.02 | 14.80 | 5.37 | 38.15 | 5.60 | 41.06 |
| **MASS (Qwen)** | 8.16 | 41.74 | 5.90 | **33.43** | 6.50 | **43.49** | 4.65 | **33.32** | 7.62 | **62.87** | 6.28 | **55.88** |
| **MASS (GPT-OSS-120B)** | **8.24** | **41.96** | **5.91** | 33.28 | **6.62** | 41.96 | 4.63 | 30.19 | **7.66** | 61.56 | **6.43** | 54.29 |

| Method | SSE50 | | | | CSI 300 | | | | CSI A500 | | | |
|---|---|---|---|---|---|---|---|---|---|---|---|---|
| | RIC | RICIR | IC | ICIR | RIC | RICIR | IC | ICIR | RIC | RICIR | IC | ICIR |
| **Experiments on data leakage concern (The first quarter of 2025)** | | | | | | | | | | | | |
| Proxy Indicator (Diether et al., 2002) | 1.46 | 10.60 | 1.51 | 9.89 | 1.52 | 10.37 | 2.01 | 14.28 | 1.04 | 9.75 | 0.98 | 9.97 |
| LightGBM (Ke et al., 2017) | 1.66 | 12.35 | 1.58 | 11.73 | 1.59 | 8.79 | 1.85 | 11.97 | 1.77 | 12.84 | 1.58 | 12.60 |
| FactorVAE (Duan et al., 2022) | 3.59 | 21.61 | 5.41 | 31.14 | 3.37 | 29.67 | 2.65 | 26.96 | 4.32 | 40.87 | 4.01 | 36.70 |
| HireVAE (Wei et al., 2023) | 3.68 | 21.52 | 5.44 | 30.15 | 3.47 | 31.58 | 2.61 | 27.93 | 4.24 | 42.69 | 3.91 | 36.94 |
| DTML (Yoo et al., 2021) | 3.53 | 20.94 | 5.28 | 28.77 | 3.39 | 28.86 | 2.54 | 27.78 | 4.06 | 41.80 | 3.75 | 35.22 |
| MASTER (Li et al., 2024b) | 3.70 | 21.38 | 5.49 | 30.26 | 3.46 | 29.74 | 2.58 | 28.47 | 4.13 | 45.52 | 3.89 | 36.67 |
| SEP (Koa et al., 2024) | 3.65 | 20.92 | 5.47 | 29.99 | 1.45 | 10.06 | 0.84 | 9.76 | 4.25 | 46.31 | 3.96 | 38.75 |
| FinCON (Yu et al., 2024) | 3.97 | 22.03 | 5.68 | 31.42 | 1.54 | 13.98 | 0.80 | 10.72 | 4.81 | 48.25 | 4.34 | 43.96 |
| TradingAgents Xiao et al. (2025b) | 4.02 | 21.94 | 5.71 | 31.99 | 3.63 | 29.80 | 2.97 | 30.63 | 4.86 | 48.95 | 4.20 | 43.94 |
| **MASS (Qwen)** | 4.50 | 24.41 | 6.12 | **38.33** | 3.91 | 37.44 | 3.36 | 34.56 | 5.19 | **56.17** | 4.66 | **48.82** |
| **MASS (GPT-OSS-120B)** | **4.56** | **24.56** | **6.31** | 37.98 | 3.75 | 35.86 | 3.31 | 33.80 | **5.27** | 54.72 | **4.68** | 46.05 |

**Complexity:** Although the total time complexity for a historical simulation is $O(n_{\text{type}} \times n_{\text{inv}} \times T)$, in a live trading scenario, the daily cost is only $O(n_{\text{type}} \times n_{\text{inv}})$. This is because we can store the latest agent distribution snapshot and update it with the newly arriving data stream. To ensure MASS's reproductivity, a detailed analysis of time and computational costs is provided in Appendix A.3.7.

**Dataset and Stock Pools:** While prior studies Koa et al. (2024); Zhang et al. (2024c); Xiao et al. (2025b) provide valuable insights, their evaluations often focus on US markets during stable bull periods Nasdaq (2025). To test model robustness in a more volatile context, we introduce a new dataset from the Chinese A-share market. Our dataset covers the entirety of 2023, a period marked by high volatility and two major shifts, thus offering a challenging benchmark. To foster further research, we have open-sourced one of our dataset. The data covers three key indices: *SSE 50* (China Securities Index Co., 2020), *CSI 300* (China Securities Index Co., 2023), and *ChiNext 100* (Shenzhen Securities Information Co., 2019). Details about the construction of our dataset are in Appendix A.2. Furthermore, to validate MASS's generalizability across different kinds of assets, we conduct experiments on Nasdaq 100 (Nasdaq, 2025) and S&P 500 (NYSE, 2025) collected from Microsoft Qlib (Yang et al., 2020) and Yahoo Finance within the same date coverage.

**Baselines:** We compare MASS with various baselines across different categories: a traditional proxy indicator (Diether et al., 2002); a machine learning model, LightGBM (Ke et al., 2017); factor-based models: FactorVAE (Duan et al., 2022) and HireVAE Wei et al. (2023); deep learning models, DTML (Yoo et al., 2021) and MASTER (Li et al., 2024b); and three SOTA agent-based methods,

SEP (Koa et al., 2024), FINCON (Yu et al., 2024), and TradingAgents (Xiao et al., 2025b). While Mars (Li et al., 2025a) is relevant, a direct comparison is not possible because their model weights are still under review [2]. Further details about baseline descriptions and our implementations are provided in Appendix A.4.

**Metrics:** We use four standard metrics to assess both correlation and consistency: the Information Coefficient (IC) and Rank Information Coefficient (RIC) quantify Pearson and Spearman correlations between predicted ($Signal$) and actual returns ($r$), respectively. Their stability is measured by the Information Coefficient Information Ratio (ICIR) and Rank Information Coefficient Information Ratio (RICIR), defined as $\mathbb{E}[IC]/\text{Std}(IC)$ and $\mathbb{E}[RIC]/\text{Std}(RIC)$. Besides, to ensure the robustness of our evaluation process, we incorporate more metrics in Appendix A.5.

**Experiment-Specific Settings:** We implement MASS using *Qwen2.5 72B Instruct* (Qwen et al., 2025) as the primary backbone, and also employ *GPT-OSS-120B* (OpenAI et al., 2025) to validate its sensitivity to different LLMs. For the main experiments (Table 1), we set the number of agent types $n^{\text{type}} = 16$ and investors per type $n^{\text{inv}} = 32$. The candidate pool size $n^{\text{sel}}$ is 20 for SSE 50 and ChiNext 100, and 30 for the larger CSI 300. The aggregation weight $\alpha$ is set to 0.5 for SSE 50 and CSI 300, but adjusted to 0.2 for the ChiNext 100 growth market to place greater emphasis on disagreement factors ($\sigma_s$), which are more predictive when valuations disconnect from fundamentals. Our backward optimization uses simulated annealing (SA) (Kirkpatrick et al., 1983) with an initial temperature of 40, a cooling rate of 0.95, a maximum of 100 iterations, and a look-back window $\omega_{\text{opt}}$ of 5. The data leakage experiment specifically evaluates the model on data from Q1 2025, a period after the LLMs' knowledge cutoff, across the SSE 50, CSI 300, and the new CSI A500 index (China Securities Index Co., 2024). Unless otherwise specified, experiments are conducted on the CSI 300 pool; settings for experiments on US markets are identical to those for the CSI 300.

### 4.1 RESULTS AND ANALYSIS

#### 4.1.1 MAIN EXPERIMENTS

Table 1 presents the primary comparison against baselines. The key observations are twofold. First, MASS achieves the best performance across all metrics and stock pools, consistently outperforming the next-best methods (TradingAgents, FactorVAE, HireVAE, SEP, MASTER, and FinCON). Second, we observe that while agent-based methods like SEP and FINCON perform reasonably on smaller pools, their effectiveness diminishes significantly on the larger CSI 300. Our analysis indicates this is because their self-reflection mechanisms, which require processing extensive historical results in-context, face comprehension and decision-making challenges with an increasing number of stocks. MASS avoids this bottleneck as its architecture does not require any single agent to process vast global information, demonstrating enhanced scalability.

Besides, we conduct two extensive experiments to validate MASS's generalizability. Firstly, to validate MASS's generalizability across different kinds of assets, we present the comparison between MASS and all baselines on the US stock market within the same date coverage in Appendix A.5. The results show that MASS still exhibits enhanced performances. Secondly, to validate MASS's generalizability between different LLM backbones, we use GPT-OSS-120B OpenAI et al. (2025) as a substitute. As is shown in Table 1, MASS shows a slight and acceptable performance difference between the two LLM backbones.

#### 4.1.2 EXPERIMENTS ON DATA LEAKAGE CONCERN

To demonstrate that MASS's performance is not attributable to data memorization, we evaluate it on unseen market data from Q1 2025. This period is subsequent to the knowledge cutoffs of the large language models used, as both Qwen 2.5 and the GPT-OSS series [3] were trained on data from before 2025. As shown in the lower part of Table 1 and Table 8 in the Appendix, MASS maintains significant effectiveness on both unseen data from existing indices (SSE 50, CSI 300 in Q1 2025), a completely new stock pool (CSI A500) and different asset classes (Nasdaq 100 and S&P 500). The results provide strong evidence that the model's success stems from its methodological framework rather than prior knowledge.

---

[2] https://github.com/microsoft/MarS

[3] https://platform.openai.com/docs/models/gpt-oss-120b

### 4.1.3 BACKTESTING EXPERIMENTS

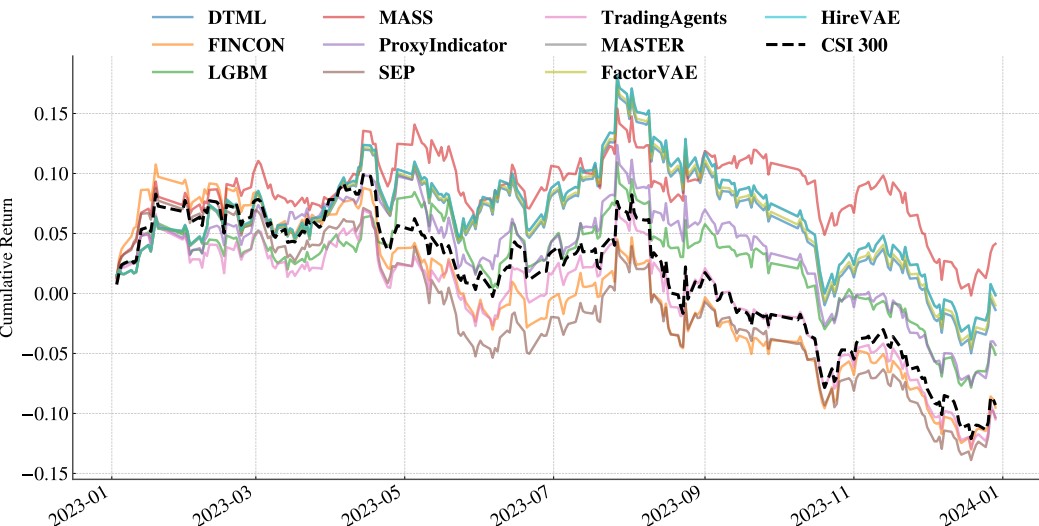

Figure 2: Backtesting on the CSI 300 Stock Pool compared with Baselines and the CSI 300 Index.

Figure 2 translates the statistical metrics into a practical financial outcome via backtesting. The plot of cumulative excess returns shows that MASS not only generates substantially higher returns than the baselines and the CSI 300 index but also maintains significantly lower drawdowns. This result highlights MASS's dual advantages in both profitability and risk control, underscoring its real-world applicability. Backtesting implementation details are in Appendix A.6.

### 4.1.4 SCALING EXPERIMENTS

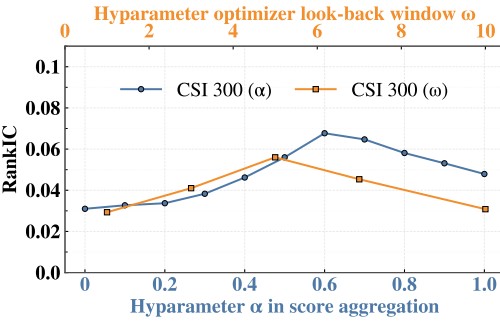

Figure 3: MASS exhibits a moderate sensitivity to changes in hyperparameters.

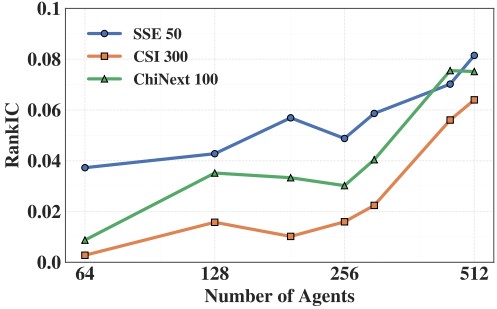

Figure 4: As the number of agents increases exponentially, MASS is able to obtain even more refined market information.

To verify the multi-agent scaling effect, we investigated the performance of MASS as we exponentially increased the number of agents ($n^{\text{type}} \times n^{\text{inv}}$) while keeping other parameters fixed. The results in Figure 4 show a clear, approximately linear growth in the RankIC metric as the total number of agents increases. This confirms that by simulating more agents, MASS is able to capture more refined market information, leading to better investment decisions. To the best of our knowledge, we are the first to explore this scaling effect in multi-agent simulation for portfolio construction, expanding the agent count up to 512.

To further investigate the potential boundary of MASS's scaling effect and the trade-off between performance and system complexity, we expand a larger number of agents(1024 & 1536) on the SSE 50 Index, which is shown in the Appendix A.3.4.

### 4.1.5 ABLATION STUDIES

Table 2: Ablation study results for CSP, PMD, BO, MDH, and an investigation of MASS , which daily updates the candidate stock pool, called MASS(DU). The best performance is indicated in **bold**. The EMCL refers to the inability to operate when exceeding the maximum context length of the LLM. All results are performed in percent.

| Method | SSE 50 | | | | CSI 300 | | | | Chi Next 100 | | | |
|---|---|---|---|---|---|---|---|---|---|---|---|---|
| | RIC | RICIR | IC | ICIR | RIC | RICIR | IC | ICIR | RIC | RICIR | IC | ICIR |
| w/o CSP | 1.65 | 11.19 | 1.67 | 11.73 | | EMCL | | | | EMCL | | |
| w/o PMD | 5.25 | 29.75 | 3.43 | 21.10 | 2.57 | 33.38 | 2.23 | 30.64 | 2.26 | 17.16 | 2.99 | 22.70 |
| w/o BO | 0.76 | 4.75 | -0.13 | -8.44 | 0.36 | 5.36 | 0.41 | 6.69 | 2.88 | 19.43 | 3.12 | 22.03 |
| w/o MDH | 6.28 | 32.68 | 3.85 | 25.39 | 4.65 | 31.03 | 2.98 | 27.86 | -3.12 | -28.93 | -2.46 | -26.44 |
| **MASS(DU)** | 8.03 | 41.68 | 5.79 | **33.52** | 6.48 | 42.86 | 4.52 | 32.95 | **7.65** | **63.02** | **6.29** | **55.91** |
| **MASS** | **8.16** | **41.74** | **5.90** | 33.43 | **6.50** | **43.49** | **4.65** | **33.32** | 7.62 | 62.87 | 6.28 | 55.88 |

Table 2 presents the results of ablating four key design choices and our variant of our proposed MASS:

- **w/o CSP (Candidate Stock Pool):** Removing this component causes the model to fail on larger indices due to exceeding the LLM's context length (EMCL). This confirms that CSP is essential for the system's scalability.

- **w/o PMD (Provide Macro Data):** Removing macroeconomic data leads to a significant performance drop, as agents lack the context to make diverse, timely decisions, thus reducing system randomness and adaptability.

- **w/o BO (Backward Optimization):** This is the most critical ablation study. Disabling the optimization process in Equation 5 causes performance to collapse, yielding near-zero or negative IC values. This proves that the end-to-end, adaptive learning of agent distribution is the core mechanism driving MASS's success.

- **w/o MDH (Market Disagreement Hypothesis):** Relying solely on consensus led to a major performance drop and was even counterproductive on the ChiNext index, demonstrating the importance of our theory-grounded aggregation method.

- **MASS(DU) (Daily Updated Candidate Stock Pool):**In Section 3.1.1, we construct each agents' a static candidate stock pool. To confirm the robustness of MASS and eliminate the possible impact of this pre-defined set, we also test a variant which updates each agent's candidate stock pool on each trading day, finding its impact negligible. This suggests that the key is the partitioned view, not whether the view is static or dynamic.

### 4.1.6 STABILITY AND AGENT DISTRIBUTION VISUALIZATION EXPERIMENT

Figure 5 visually demonstrates the adaptability of MASS. The background color tracks the temporal evolution of the agent distribution throughout 2023, with the left y-axis representing the proportion of different agent types. The right y-axis indicates the cumulative return, which is used to plot the performance of our MASS (blue line) and the CSI 300 Index (black dashed line). The orange line illustrates the excess cumulative return of MASS compared to the CSI 300 benchmark. The two major market shifts in February (rebound to consolidation) and August (consolidation to decline) are marked as A and B. During both transitions, MASS's backward optimization mechanism adapted the agent distribution to align with the new market style. This rapid adaptation enabled MASS to maintain stable excess returns over the CSI 300 index, even during periods of high volatility.

To provide a deeper understanding of our model's internal dynamics and inference stability, we present two case studies in Appendix A.3.5. The first offers a micro-level analysis of agent interactions during a market regime shift, showing how our backward optimization mechanism adjusts agent weights to foster a new investment consensus that enhances returns. The second case study addresses the strategic consistency of MASS, demonstrating that despite the stochastic nature of LLMs, the model generates robust and consistent stock popularity patterns across independent inference runs.

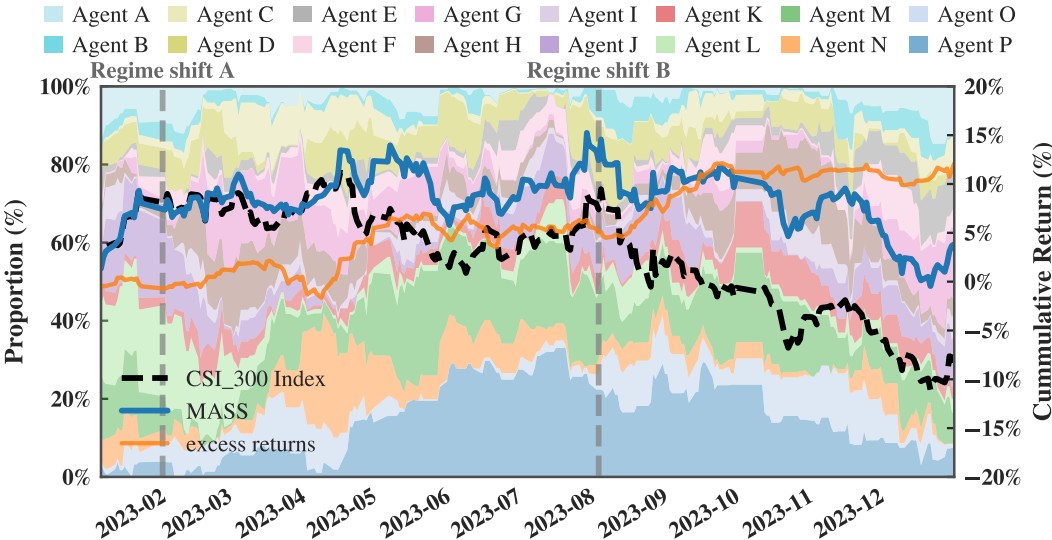

Figure 5: The distribution of agents in MASS swiftly adapts to changes in market styles (A and B), allowing it to consistently achieve stable excess returns compared to the CSI 300 Index.All legends placed in the top of this figure denote an agent type.

#### 4.1.7 PARAMETER SENSITIVITY EXPERIMENTS

To investigate the sensitivity of MASS to its parameters, we analyzed four hyperparameters: the score aggregation weight $\alpha$ (Equation 4) the optimizer look-back window $\omega_{opt}$, the optimizer iteration times, and the optimizer cooling rate. The parameter $\alpha$ manages the balance between disagreement and consensus components in portfolio construction, while $\omega_{opt}$ influences information capacity—too short a window limits it, whereas too long a duration hinders regime adaptation. The experimental results are presented in Figure 3, and the detailed analysis of the other two hyperparameters' impact on system performance is provided in the Appendix A.3.6.

We observe that although adjustments to these four hyperparameters lead to slight variations in system performance, these changes are within acceptable limits. This indicates that MASS exhibits a moderate sensitivity to parameter changes.

## 5 CONCLUSION

In this paper, we introduce MASS, a multi-agent scaling simulation framework designed for portfolio construction. MASS leverages large-scale agent simulations and a backward optimization process to achieve a comprehensive understanding of market dynamics. This approach offers various advantages, including enhanced scalability, robustness, and the ability to generate stable excess returns.

In the future, we anticipate that the paradigm established by MASS will extend beyond investment portfolio management to encompass a wider range of tasks, such as supply chain optimization, agricultural decision-making, and weather prediction.

## 6 ACKNOWLEDGEMENTS OF LLM USAGE

We declare that the use of LLMs during the preparation of this manuscript was strictly limited to language-related assistance, such as sentence refinement and grammatical correction. All substantive content was independently authored by the authors and rigorously reviewed and verified following any LLM-assisted modifications. Detailed experimental settings are provided in the Experiments section of this paper. No other reliance on LLMs is involved in this work.

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

## A  APPENDIX

### A.1  HIGH-LEVEL WORKFLOW OF MASS

---
**Algorithm 1:** MASS: Online Learning Framework

---
**Input:** Multi-modal stock features $\mathcal{X}$, macroeconomic data $\mathcal{M}$, historical stock returns $\mathbf{Y}$, number of agent types $n^{\text{type}}$, agents per type $n^{\text{inv}}$, look-back window $\omega_{\text{opt}}$, trading days $T$

**Output:** Daily investment portfolio $\mathbf{P}$

1 Initialize agent type distribution $\mathbf{d}_0 \leftarrow [\frac{1}{n^{\text{type}}}, \ldots, \frac{1}{n^{\text{type}}}]^\top$;

                                  // Uniform initial distribution

2 Initialize all agents $(i, k)$ for $i \in \{1, \ldots, n^{\text{type}}\}$, $k \in \{1, \ldots, n^{\text{inv}}\}$;

3 **for** *each trading date $j \in T$* **do**

    // − Forward Propagation: Generate signal for day j −

4     **for** *agent type $i = 1$ to $n^{type}$* **do**

5         **for** *agent $k = 1$ to $n^{inv}$* **do**

6             Generate investment strategy $\text{Strategy}_{i,j}$ using Eq. 1;

7             Agent $(i, k)$ selects stocks $\text{Codes}_{i,k,j}$ via Eq. 2;

8     Aggregate agent decisions to compute $\text{Signal}(s, j)$ for all stocks $s$ via Eqs. 3, 4 using distribution $\mathbf{d}_{j-1}$;

    // − Portfolio Construction for day j −

9     Construct portfolio $\mathbf{P}_j$ from $\text{Signal}(:, j)$ using a Top-$k$ strategy;

    // − Backward Optimization: Update distribution for day j+1 −

10     Optimize distribution $\mathbf{d}_j$ using historical data up to day $j$ via Eq. 5;

11 **return** *Sequence of daily portfolios $\{\mathbf{P}_j\}_{j \in T}$*;

---

## A.2 DATASET DETAILS

### A.2.1 STOCK POOL DETAILS

- **SSE 50:** This index includes the 50 largest and most liquid stocks on the Shanghai Stock Exchange, mainly large state-owned enterprises and industry leaders. It is stable and blue-chip, suitable for risk-averse and long-term investors focusing on defensive strategies.
- **CSI 300:** Comprising the top 300 stocks from the Shanghai and Shenzhen markets, this index covers diverse industries and company sizes, offering broad market representation. It is ideal for investors seeking diversification and medium- to long-term returns.
- **ChiNext 100:** Featuring 100 stocks from the Shenzhen ChiNext Market, this index focuses on high-tech and innovative firms. Known for its growth potential and higher volatility, it suits investors with high-risk tolerance and those interested in technology sectors.
- **CSI A500:** This index selects 500 leading stocks from A-shares, covering all 35 CSI secondary industries and 91 out of 93 tertiary industries. It emphasizes sector-balanced exposure, ESG screening, and inclusion of innovative "New Quality Productivity" sectors (e.g., IT, industrials, healthcare). With strong profitability (71% of A-share net profits) and low valuation (14.16x P/E), it serves as a "China S&P 500" for diversified core-asset allocation and long-term growth strategies.

### A.2.2 DATASET CONSTRUCTION DETAILS

We construct our dataset with individual stock data and macroeconomic data.

**Individual stock data**

- **News**: Stock news is collected from various data sources. We use their titles and summaries as a substitute.
- **Financial Report**: Financial Report is collected from Wind API. We use their titles and summaries as a substitute.
- **E/P_TTM**: The inverse of the P/E ratio (E/P) indicates the earnings yield, showing the percentage of profit generated per dollar invested in the stock.
- **B/P_TTM**: Inverse of P/B (B/P) indicates the book yield, showing the return on book value per dollar invested.
- **S/P_TTM**: The inverse of the price-to-sales ratio (S/P) reflects the sales yield, quantifying the amount of sales revenue generated for each dollar invested in the company. A higher value indicates greater efficiency in converting investment into sales.
- **CF/P_TTM**: Inverse of P/CF (CF/P) shows the cash flow yield, representing cash flow generated per dollar invested.
- **Log-orthogonalized E/P**: Log-orthogonalized version of E/P, removing some kind of cap basis.Log-orthogonalized version of E/P, removing some kind of cap basis.
- **Log-orthogonalized B/P**:Log-orthogonalized version of the book-to-price ratio, which accounts for and removes certain capitalization effects, thereby isolating the information content of B/P independent of market capitalization.
- **Log-orthogonalized CF/P**:The log-orthogonalized version of the cash flow-to-price ratio, which is employed to control for capitalization influences, ensuring that the ratio captures the true predictive power of cash flow relative to price.
- **Log-orthogonalized S/P**:Log-orthogonalized version of S/P, removing some kind of cap basis.
- **EBITDA/EV**: Measures a company's return on enterprise value, indicating operating earnings (EBITDA) generated per dollar of EV.
- **ROE** : ROE measures profitability by indicating how much net income is generated for each dollar of shareholders' equity. Higher values signify more effective utilization of equity capital to generate earnings.
- **ROE stability**: TS_Mean(ROE, 8) / TS_Std(ROE, 8), measuring both absolute value and stability of ROE.

- **ROA stability**: TS_Mean(ROA, 8) / TS_Std(ROE, 8), measuring both absolute value and stability of ROA.
- **Dividend yield**: Dividend yield indicates annual dividends received per dollar invested, expressed as a percentage of the stock price.
- **Log-orthogonalized dividend yield**: Log-orthogonalized version of dividend yield, removing some kind of cap basis.
- **Dividend yield incl repo & mjrholder trans**: Dividend yield including stock repurchasing and major holder trading.
- **Revenue TTM YoY growth rate**: Measures the percentage change in trailing twelve months' revenue compared to the same period last year.
- **Net profit TTM YoY growth rate**: Measures the percentage change in trailing twelve months' net profit compared to the same period last year.
- **Non-GAAP net profit YoY growth rate**: Indicates the percentage change in non-GAAP net profit compared to the same period last year.
- **Interday volatility**: The price fluctuation range of a stock across trading days.
- **Liquidity**: Weighted average of monthly, quarterly, and yearly turnover ratios.
- **Residual volatility**: Residual volatility measures the unexplained variability in a security's returns after accounting for market or factor influences, indicating idiosyncratic risk.
- **Stock Base data**: The open, high, low, close, volume, and value data of individual stocks on a daily timeframe. (forward-adjusted)
- **industry index return**: One-day return of holding the sector's constituent stocks.
- **Price-volume feature**: Various features extracted from Alpha 158 (Yang et al., 2020) based on price and volume.

**Macroeconomic data**

- The latest 1-year loan prime rate.
- The latest month China CPI YOY growth rate.
- The latest yield of China ten ten-year government bonds.
- The latest PE and PE quantile of the CSI 300 index.

### A.3 More details about Mass

#### A.3.1 More details of Market Disagreement hypothesis

Market disagreement describes heterogeneous investor beliefs that drive trading activities. The market disagreement hypothesis posits that such divergence systematically distorts security valuations: when optimistic investors dominate trading while pessimists face short-selling constraints, securities become overpriced and exhibit lower future returns (Miller, 1977). This theory establishes disagreement as a persistent market friction that generates predictable return patterns, with empirical studies confirming that **high-disagreement stocks consistently underperform consensus-driven counterparts** (Diether et al., 2002; Sadka & Scherbina, 2007).

#### A.3.2 The Design of Investor Initialization

**System & User Prompts**

**System Prompt**
You are a helpful assistant. Make sure you carefully and fully understand the details of the user's requirements before you start solving the problem.

**User Prompt**
Give the following input data:
1. Input time-series data column name and their descriptions in JSON format(textual data example).
2. latest macroeconomic and market insights. Please try to analyze and summarize an abstract investing style description.

```
The output format is a json. The specific format of the output JSON is:
{ "Outline": "The outline and general description for investment style within 50 words.
The outline is a summarization about your investing strategy and your insights into
the subsequent trend of the stock market, without any details below.",
"Details": { "Risk Appetite": "conservative | moderate | moderately conservative |
moderately aggressive | aggressive",
"Holding Period": "one day | about one week | about one month | about half a year |
more than one year",
"Strategy Consistency": [0, 1] (Refers to the investor's ability to adhere to and execute
their investment strategy with persistence and coherence, regardless of short-term
market fluctuations or emotional influences. Higher number means high consistency",
"Rationality": [0, 1] (Refers to whether the investor's decision-making process is based
on logic, data, and long-term objectives rather than emotions, biases, or short-term
market noise. Higher number means high rationality",
"StockPoolSelector": "Specify what kind of preference you'd like to construct your
watchlist stocks. The possible preferences are:
1. RandomStockSelector: Randomly construct your watchlist.
2. IndustryEqualStockSelector: Construct a stock pool with balanced distribution across
industries.
3. MVEqualStockSelector: Construct a stock pool with balanced distribution across
market capitalizations.
4.IndustryBasisStockSelector: Prefer stocks from specific industries and output the
preferred industries. The result is presented in a list format.",
"Others": "Extra information about your investing strategy, maybe correlated with
latest market and macroeconmic information and others. No more than 30 words." } }
{examples}
Input data:
E/P,B/P,CF/P, S/P,Log-orthogonalized E/P,Log-orthogonalized B/P,Log-orthogonalized
CF/P,Log-orthogonalized S/P,
Macro data:
The latest 1-year loan prime rate is 3.45. The latest month China CPI YOY growth rate
is -0.5. The latest yield of China's ten-year government bonds is 2.6733%, while the
yield has increased 0 BP over the past one day, increased -4 BP over the past one month,
and increased -21 BP over the past half a year. The latest CSI_300 PE is 10.9478,
and the current PE ratio of the CSI 300 is at the 5.4th percentile over the past 5
years(0 indicates most undervalued, and 100 indicates most overvalued). The latest
market sentiment index got a 0.63% return.
Your investing style:
{'Outline': 'A value-oriented investment approach focusing on fundamentally strong
companies with a long-term perspective, leveraging current market undervaluation and
stable economic indicators to build a diversified portfolio.',
'Details': {'Risk Appetite': 'moderate', 'Holding Period': 'more than one
year', 'Strategy Consistency': '0.85', 'Rationality': '0.9', 'StockPoolSelector':
'IndustryEqualStockSelector', 'Others': 'Leverage low CPI and undervalued CSI 300
PE for potential upside.'}
(END_OF_EXAMPLES)
Input data: {input_data}
Macro data: {macro_data}
Your investing style:
```

### A.3.3 THE DESIGN OF INVESTMENT STRATEGY EXECUTION

**User Prompts**

**User Prompt**
```
Giving following
1. Input data in table format and their descriptions in JSON format.
2. investing style to make investment decisions in JSON format.
Please output {num_stocks} stocks you tend to invest in. The result is in JSON format,
key is "Stock", and value is a list containing the stock code. Please make sure:
1. You output legal stock code. The stock code is legal if and only if it is in the
input data "Stock" list.
```

2. The number of stock codes is correct, actually equal to {num_stocks}. Here is an example.
For stock_nums in investing instructions, we use 3 in this example. Input Data for investing decision:
1. **Input Data Description**:
{"E/P": "The inverse of the P/E ratio (E/P) indicates the earnings yield, showing the percentage of profit generated per dollar invested in the stock.",
"B/P": "Inverse of P/B (B/P) indicates the book yield, showing the return on book value per dollar invested.",
"S/P": "Inverse of P/S (S/P) reflects the sales yield, showing sales generated per dollar invested.",
"CF/P": "Inverse of P/CF (CF/P) shows the cash flow yield, representing cash flow generated per dollar invested.",
"Log-orthogonalized E/P": "Log-orthogonalized version of E/P, removing some kind of cap basis.",
"Log-orthogonalized B/P": "Log-orthogonalized version of B/P, removing some kind of cap basis.",
"Log-orthogonalized CF/P": "Log-orthogonalized version of CF/P, removing some kind of cap basis.",
"Log-orthogonalized S/P": "Log-orthogonalized version of S/P, removing some kind of cap basis.",
"EBITDA/EV": "Measures a company's return on enterprise value, indicating operating earnings (EBITDA) generated per dollar of EV."}
2. **Investing Style**:
{"Outline": "A value-driven investment approach focusing on stocks with strong fundamentals, undervalued valuations, and consistent cash flows over the long term.",
"Details": { "Risk Appetite": "Moderately conservative", "Holding Period": "More than one year", "Strategy Consistency": "0.85", "Rationality": "0.9", "StockPoolSelector": "MVEqualStockSelector" }}
3. **Input data**:
',Stock,Date,E/P,B/P,CF/P,S/P,
Log-orthogonalized E/P,Log-orthogonalized B/P,Log-orthogonalized CF/P,
Log-orthogonalized S/P, EBITDA/EV,
965494,000858,20190102, 0.06295366,
0.30744636,0.038947526,0.19324197,
-4.032941,-1.1295723,3.594055,
-1.2754831,0.124886042941460,
002594,20190102,0.020888906,
0.37708813,0.09185906,0.9017491,-4.038043,
-0.6966869,5.084233,0.3152281,0.09258402716,
600519,20190102,0.042301364,0.13605072,
0.036664255,0.09038502,-7.6968794,-2.2439895,
1.2049837,-2.2207088,0.0797575348104294,
600900,20190102,0.066111766,0.4052357,0.1183,
0.15322393,-5.3881683,-1.0025798,3.743841,
-1.5840118,0.1050353948267292,601012,
20190102,0.062190603,0.30756927,0.032795224,
0.41643697,-0.72993636,-0.7708632,
5.801872,-0.31826368,0.0887390158431868,
601288,20190102,0.16604953,1.2584949,
0.12149128,0.4757528,-7.5973797,-0.1158539,
1.556502,-0.6272717,0.059454748665067,
601888,20190102,0.02850359,0.1358013,
0.034710173,0.35662726,-3.433404,-1.7193639,
4.34933,-0.59344673,0.0511954139068489,
603259,20190102,0.024971908,0.12955885,
0.018961666,0.10751114,-2.9358995,
-1.8100101,4.314365,-1.7471998,0.04303389'
**LLM output**:
{'Stock': ['000858', '600900', '601288']}
Note that in this example, we ask LLM to output 3 stocks. However, in real scenarios, you should follow the "num_stocks" args in the instruction.
(END OF EXAMPLES)

```
{input_data}
```

### A.3.4 More discussions on the scaling effect

We investigate the scaling properties of MASS by varying the number of agents (NOA). The results, presented in Table 5, indicate that performance improves as the NOA increases to 1024, but plateaus thereafter. Based on publicly available fund performance data [4], an annualized excess return of 12 - 15% over large-cap benchmarks such as the SSE 50 is considered near the optimal ceiling for strategies that do not rely on high-frequency features or market timing (MASS reaches 12.14% when NOA is set to 512). We therefore cautiously suggest that the performance of MASS may approach its upper bound when the NOA is configured between 512 and 1024.

### A.3.5 Case Study

Firstly, to provide a micro-level analysis of agent interactions, we conduct a case study on two representative stocks from the CSI 300 index: China Shenhua Energy (601088.SH), visible to Agent O, and Kweichow Moutai (600519.SH), visible to Agent D. During the market regime shift B (identified in Figure 5), different agents initially propose strategies based on their intrinsic preferences—for instance, Agent A favors dividend assets while Agent D favors consumer stocks. Following the shift, our backward optimization mechanism detects a potential change in market style based on lookback window performance. Consequently, the mechanism increases the allocation to Agent O and reduces the allocation to Agent D. This reallocation fosters a higher consensus on China Shenhua over Kweichow Moutai, ultimately contributing to the enhanced returns of MASS.

---

**Investment agent examples**

**Agent O**
```
{ 'Outline': 'A value-oriented investment approach focusing on fundamentally strong
companies with a long-term perspective, preferring assets with low valuation and
delivering stable and long-term return for holders',
'Details': Ŕisk Appetite': 'moderate',
'Holding Period': 'more than half a year',
'Strategy Consistency': '0.85',
'Rationality': '0.9',
'StockPoolSelector': 'IndustryEqualStockSelector',
'Others': 'A declining CPI growth rate and the government bond yields suggest that
dividend assets may outperform.',
'Visible stocks': ['601088.SH', '600030.SH', '002594.SZ', ...]
}
```
**Agent D**
```
{ 'Outline': 'A value-oriented investment approach focusing on companies with stable
and strong cash flows, preferring assets with lower valuation and higher profit
quality',
'Details': {'Risk Appetite': 'moderate',
'Holding Period': 'more than half a year',
'Strategy Consistency': '0.85',
'Rationality': '0.9',
'StockPoolSelector': 'IndustryEqualStockSelector',
'Others': 'Macroeconomic stimulus policies may lead to a recovery in consumption.',

'Visible stocks': ['600519.SH', '603259.SH', '000858.SZ', ...]}

}
```

---

Secondly, we provide another case study to explore how the stochastic process in the LLM generation might influence MASS's inference outcomes in a strategy pattern. We define the popularity of a stock $s$ on a given day $j$ as the total number of agents selecting it, weighted by the agent distribution, just like Equation 3a. Subsequently, on each trading day, we derive a popularity probability distribution across our stock universe. This is achieved by normalizing the popularity of each individual stock by the cross-sectional sum of all popularities for that day.

---

[4]https://www.simuwang.com/

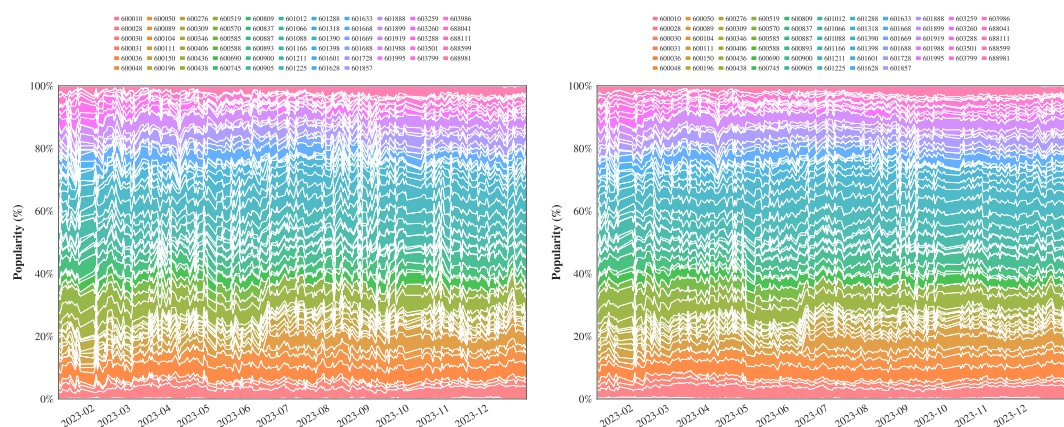

(a) All stocks aggregated popularity among all agents during the full 2023 on the SSE 50 Index. The first inference trail.

(b) All stocks aggregated popularity among all agents during the full 2023 on the SSE 50 Index. The second inference trail.

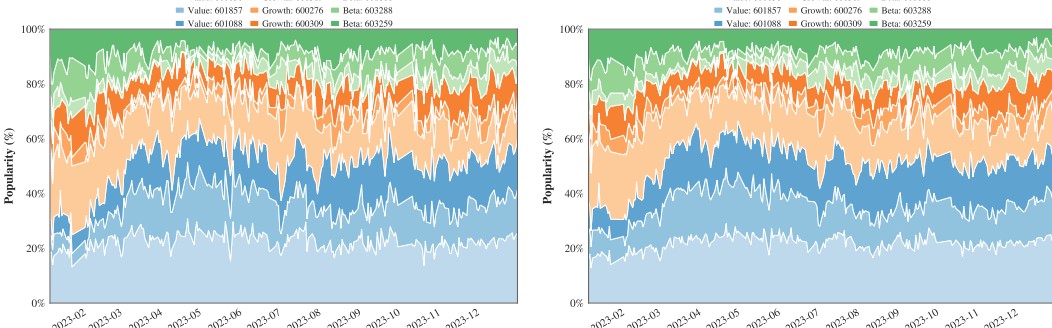

(c) 9 stocks with 3 different styles aggregated popularity among all agents during the full 2023 on the SSE 50 Index. The first inference trail.

(d) 9 stocks with 3 different styles aggregated popularity among all agents during the full 2023 on the SSE 50 Index. The second inference trail.

Figure 6: Comparison of investor popularity under two independent inference trails. For improved readability, the popularity term is transformed into a probability distribution. Figures (c) and (d) show magnified views of specific areas in Figure (a) and (b), with the data in these sections subsequently converted back into probability distributions.

To illustrate this, we conducted two independent inference runs on the SSE 50 Index. The resulting evolution of these probability distributions over time from each run is visualized in Figure 6a and Figure 6b, respectively. We observe no significant variation in stock popularity distributions between the two runs.

However, the large number of stocks in Figures 6a and 6b makes detailed inspection challenging. For a more granular analysis, we select a representative subset of nine stocks spanning diverse industries and styles (value, growth, and beta), according to the China Securities Index (CSI) classification methodology[5]. Detailed information on these stocks is provided in Table 3.

Their popularities are magnified and presented in Figures 6c and 6d. From these figures, we can observe subtle, yet not significant, variations between the two independent runs. More importantly, the popularity of these nine stocks follows a broadly consistent evolutionary pattern across both inference processes.

---

[5]https://www.csindex.com.cn/#/

Table 3: Details of the 9 selected A-share stocks with hierarchical style grouping.

| Style | Ticker | Company |
|---|---|---|
| **Value** | 600036 | China Merchants Bank Co., Ltd. |
| | 601857 | PetroChina Company Limited |
| | 601088 | China Shenhua Energy Company Limited |
| **Growth** | 600519 | Kweichow Moutai Co., Ltd. |
| | 600276 | Jiangsu Hengrui Pharmaceuticals Co., Ltd. |
| | 600309 | Wanhua Chemical Group Co., Ltd. |
| **Beta** | 601888 | China Tourism Group Duty Free Corporation Limited |
| | 603288 | Foshan Haitian Flavouring & Food Co., Ltd. |
| | 603259 | WuXi AppTec Co., Ltd. |

### A.3.6 MORE HYPERPARAMETER SENSITIVITY ANALYSIS

Table 4 presents the hyperparameter sensitivity of MASS's optimizer on the CSI 300. The results are twofold. The optimizer demonstrates considerable robustness to hyperparameter tuning. For the cooling rate, values within the 0.85 ~ 0.98 range yielded consistent and strong results; optimization failure occurred only upon the complete omission of the cooling schedule. Similarly, the model reliably converged with over 100 iterations, indicating a wide tolerance for this parameter. Non-convergence was observed only with an insufficient number of iterations. In light of these findings, we cautiously suggest that our optimizer is not overly sensitive to its hyperparameter settings.

### A.3.7 TIME COMPLEXITY AND TOKEN COST

For practical deployment, the operational efficiency of MASS is a key consideration. We have incorporated several strategic design choices to minimize API-related costs and ensure computational feasibility in a live trading scenario.

First, we recognize that the macroeconomic data informing the investor initialization module (Section 3.1.1) changes at a low frequency. For instance, metrics like the CPI growth rate are updated monthly, while others, such as government bond yields and market-wide PE percentiles, evolve gradually. To substantially mitigate API overhead, we therefore limit the re-invocation of this initialization module to only once per week, specifically on the first trading day.

Second, and critically, we cache the individual investment decisions generated by each agent during the forward propagation stage. Consequently, the backward optimization process—while computationally intensive in its re-aggregation of historical signals for different candidate distributions—operates entirely on these cached results. This design ensures that the backward optimization step incurs **zero additional LLM token costs**.

These optimization strategies ensure the economic viability of MASS in a real-world setting. To provide a transparent assessment of its practicality, we report the average daily computational time and API costs for a 512-agent configuration in Table 6.

### A.4 BASELINE DETAILS

- **Proxy indicators**: Various features can be used as a proxy to quantify market disagreement (Diether et al., 2002). We use the earning stability of the listed company (implemented by calculating the std of annualized ROE) as a baseline.
- **LightGBM**: A high-efficiency, leaf-wise gradient boosting decision tree framework by Microsoft Research, employing histogram-based algorithms for accelerated training and reduced memory footprint. Following (Bali et al., 2023) and Equation 4, we simulate market disagreement by constructing various LightGBM agents visible to different features.
- **FactorVAE**: FactorVAE (Duan et al., 2022) is a popular probabilistic dynamic factor model based on variational autoencoder. We use the open-source implementation [6] to implement FactorVAE.

---

[6] https://github.com/x7jeon8gi/FactorVAE

Table 4: Optimizer hyperparameter sensitivity on **CSI 300**. Metrics: RIC / RICIR / IC / ICIR. Best values in each column are recommended to be highlighted in **bold**.

| Sensitivity to Optimizer Hyperparameters | | | | |
| --- | --- | --- | --- | --- |
| **Cooling Rate** | | | | |
| **Cooling Rate** | **RIC** | **RICIR** | **IC** | **ICIR** |
| 1.00 | -0.16 | -3.58 | -0.27 | -4.99 |
| 0.98 | 5.79 | 39.68 | 4.21 | 30.81 |
| 0.95 | 6.50 | 43.49 | 4.65 | 33.32 |
| 0.90 | **6.53** | **44.82** | **4.77** | **34.06** |
| 0.85 | 5.81 | 40.12 | 4.16 | 31.13 |
| 0.80 | 4.12 | 31.90 | 3.89 | 24.58 |
| **Iteration Times** | | | | |
| **Iteration Times** | **RIC** | **RICIR** | **IC** | **ICIR** |
| 0 | 0.36 | 5.36 | 0.41 | 6.69 |
| 25 | 3.04 | 23.55 | 2.94 | 21.89 |
| 50 | 4.69 | 31.80 | 3.73 | 26.66 |
| 100 | 6.50 | **43.49** | 4.65 | **33.32** |
| 200 | **6.53** | 42.76 | **4.66** | 32.91 |

Table 5: More results: MASS scaling effect on the SSE 50 Index when the number of agents (NOA) increases. All results are in percent.

| NOA | RIC | RICIR | IC | ICIR |
| --- | --- | --- | --- | --- |
| 512 | 8.16 | 41.74 | 5.90 | 33.43 |
| 1024 | 9.25 | 43.02 | 6.27 | 34.19 |
| 1536 | 9.22 | 43.11 | 6.29 | 34.05 |

Table 6: MASS 's average time cost and api call fees on each trading day.

| Stock Pool | Time | Cost |
| --- | --- | --- |
| SSE50 | 125s | $0.679 |
| CSI 300 | 378s | $2.265 |
| Chi Next 100 | 227s | $1.192 |

- **HireVAE**: HireVAE (Wei et al., 2023) is a novel end-to-end neural factor model that can identify current market regime according to point-in-time market information, and subsequently adapt itself for better prediction.
- **DTML**: DTML (Yoo et al., 2021) is an attention-based model that exploits the correlations between stocks to make investment decisions. We use the open-source implementation [7] to implement this baseline.
- **MASTER**: MASTER (Li et al., 2024b) is a stock transformer for stock price forecasting, which models the momentary and cross-time stock correlation and guides feature selection with market information. We use the open-source implementation [8] to implement this baseline.
- **SEP**: SEP (Koa et al., 2024) utilizes a verbal self-reflective agent and A PPO that allows the LLM to teach itself how to generate explainable single stock predictions. We use the open-source link [9] to implement SEP.
- **FINCON**: FINCON (Yu et al., 2024) is a multi-agent framework for single stock price prediction and simple investment portfolio construction with conceptual verbal reinforcement.

---

[7] https://github.com/ceteris11/DTML

[8] https://github.com/SJTU-DMTai/MASTER

[9] https://github.com/koa-fin/sep

Table 7: Comparisons with baselines on more evaluation metrics. MASS outperforms almost all others across all 3 stock pools, showing impressive cumulative returns compared to the stock index. The best performance in each column is highlighted in **bold**.

| Method | SSE50 | | | CSI 300 | | | Chi Next 100 | | |
|---|---|---|---|---|---|---|---|---|---|
| **Main Experiments (Throughout 2023)** | AR | Sharpe | MDD | AR | Sharpe | MDD | AR | Sharpe | MDD |
| Proxy Indicator (Diether et al., 2002) | -2.39 | -1.22 | 14.04 | -3.60 | -1.62 | 20.57 | -20.01 | -3.24 | 24.15 |
| LightGBM (Ke et al., 2017) | -1.88 | -1.14 | 13.16 | -4.55 | -2.12 | 18.57 | -19.32 | -3.01 | 23.96 |
| FactorVAE (Duan et al., 2022) | -1.60 | -0.87 | 13.02 | -0.27 | -0.09 | 21.85 | -7.24 | -2.74 | 23.92 |
| HireVAE (Wei et al., 2023) | -1.42 | -0.95 | 12.48 | 0.96 | 0.35 | 21.70 | -7.15 | -2.69 | 23.30 |
| DTML (Yoo et al., 2021) | -1.69 | -1.08 | 12.99 | -0.33 | -0.14 | 22.34 | -8.23 | -3.20 | 24.55 |
| MASTER (Li et al., 2024b) | -1.67 | -0.92 | 12.91 | 0.79 | 0.33 | 22.05 | -7.88 | -3.17 | 24.06 |
| SEP (Koa et al., 2024) | -2.01 | -1.07 | 13.12 | -10.24 | -4.32 | 22.67 | -6.84 | -3.14 | 24.01 |
| FinCON (Yu et al., 2024) | -1.82 | -0.98 | 13.05 | -9.25 | -3.28 | 23.74 | -6.01 | -2.80 | 23.75 |
| TradingAgents (Xiao et al., 2025b) | -2.44 | -1.71 | 13.15 | -7.19 | -3.02 | 19.61 | -4.65 | -2.82 | 23.84 |
| **MASS(Qwen)** | **2.16** | 1.98 | 11.98 | **4.95** | **2.23** | **14.04** | 1.17 | **0.99** | **19.06** |
| **MASS(GPT-OSS-120B)** | 2.14 | **1.99** | **11.36** | 4.87 | 2.06 | 14.87 | **1.26** | 0.97 | 22.67 |
| Stock pool Index | -9.98 | -2.37 | 21.62 | -9.75 | -2.92 | 21.44 | -19.18 | -3.17 | 32.26 |

| Method | SSE50 | | | CSI 300 | | | CSI A500 | | |
|---|---|---|---|---|---|---|---|---|---|
| **Experiments on data leakage concern (The first quarter of 2025)** | AR | Sharpe | MDD | AR | Sharpe | MDD | AR | Sharpe | MDD |
| Proxy Indicator (Diether et al., 2002) | 0.65 | 0.16 | 5.47 | 1.98 | 0.23 | 5.94 | 1.44 | 0.20 | 6.05 |
| LightGBM (Ke et al., 2017) | 0.84 | 0.17 | 5.48 | 1.97 | 0.19 | 6.02 | 1.74 | 0.25 | 5.89 |
| FactorVAE (Duan et al., 2022) | 4.60 | 1.87 | 4.04 | 4.53 | 1.85 | 5.60 | 6.83 | 2.04 | 5.32 |
| HireVAE (Wei et al., 2023) | 4.78 | 1.92 | 4.06 | 4.81 | 2.05 | 5.01 | 7.08 | 2.20 | 5.28 |
| DTML (Yoo et al., 2021) | 4.49 | 1.70 | 4.35 | 4.55 | 1.79 | 6.06 | 6.85 | 1.93 | 6.27 |
| MASTER (Li et al., 2024b) | 5.01 | 1.98 | 3.97 | 4.78 | 1.87 | 5.45 | 6.76 | 1.97 | 4.96 |
| SEP (Koa et al., 2024) | 4.99 | 1.84 | 4.70 | 1.12 | 0.19 | 5.90 | 1.21 | 0.21 | 6.02 |
| FinCON (Yu et al., 2024) | 5.12 | 2.09 | 3.38 | 1.22 | 0.18 | 6.08 | 0.98 | 0.26 | 5.86 |
| TradingAgents (Xiao et al., 2025b) | 5.27 | 2.14 | 3.27 | 5.58 | 2.26 | **2.97** | 8.87 | 2.68 | 4.12 |
| **MASS(Qwen)** | 9.74 | **2.42** | **2.91** | 9.36 | **2.66** | 2.99 | 11.34 | **2.93** | **4.08** |
| **MASS(GPT-OSS-120B)** | **9.81** | 2.38 | 3.04 | 8.42 | 2.49 | 3.04 | **11.51** | 2.88 | 4.17 |
| Stock pool Index | -1.88 | -2.97 | 5.63 | -3.88 | -3.15 | 5.86 | -1.28 | -3.26 | 6.04 |

- **TradingAgents**: TradingAgents (Xiao et al., 2025b) is a multi-agent framework. that utilizes trading firms' collaborative dynamics to construct investment portfolios. We use the open-source link [10] to implement TradingAgents.

## A.5 FURTHER EXPERIMENT RESULTS

### A.5.1 PORTFOLIO METRICS

In this section, we provide more evaluation metrics on portfolio construction to demonstrate the superiority of MASS. These metrics are as follows:

1. **Annualized Return (AR)**
   The average annual return of the strategy, calculated by scaling the periodic return (e.g., daily, monthly) to an annual basis. It reflects the strategy's profitability over time, with the formula:
   $$R_{\text{annual}} = (1 + R_{\text{periodic}})^n - 1$$
   where $n$ is the number of periods in a year.

---

[10]https://github.com/TauricResearch/TradingAgents

Table 8: Comparison with baselines on Nasdaq-100 and S&P 500 and robustness to data leakage. We report rank information coefficient (RIC) and information coefficient (IC) along with their information ratios (RICIR, ICIR). Higher is better; best results are in **bold**. All values are in percent.

| Main experiments on Nasdaq-100 and S&P 500 (2023) | | | | | | | | |
|---|---|---|---|---|---|---|---|---|
| **Method** | **Nasdaq-100** | | | | **S&P 500** | | | |
| | RIC | RICIR | IC | ICIR | RIC | RICIR | IC | ICIR |
| Proxy Indicator (Diether et al., 2002) | 1.94 | 15.37 | 1.82 | 13.91 | 1.85 | 16.02 | 1.93 | 14.31 |
| LightGBM (Ke et al., 2017) | 2.71 | 19.90 | 2.56 | 19.34 | 2.06 | 19.84 | 2.19 | 17.83 |
| FactorVAE (Duan et al., 2022) | 3.49 | 26.05 | 3.62 | **28.95** | 3.96 | 28.34 | 3.77 | 29.64 |
| HireVAE (Wei et al., 2023) | 3.52 | 25.30 | 3.79 | 27.98 | 4.12 | 27.86 | 3.83 | 28.39 |
| DTML (Yoo et al., 2021) | 3.15 | 22.90 | 2.83 | 21.56 | 3.52 | 24.65 | 2.96 | 20.10 |
| MASTER (Li et al., 2024b) | 3.38 | 23.62 | 2.98 | 21.49 | 3.27 | 25.93 | 3.09 | 22.53 |
| SEP (Koa et al., 2024) | 3.40 | 22.99 | 3.26 | 23.85 | 1.38 | 11.82 | 0.82 | 7.81 |
| FinCON (Yu et al., 2024) | 3.46 | 23.81 | 3.24 | 24.77 | 1.24 | 10.27 | 0.68 | 8.64 |
| TradingAgents (Xiao et al., 2025b) | 3.63 | 27.36 | 3.85 | 28.29 | 4.07 | 31.28 | 3.89 | 27.94 |
| **MASS** | **4.27** | **31.05** | **3.94** | 28.90 | **4.31** | **31.45** | **3.95** | **28.68** |
| Data leakage experiments on Nasdaq-100 and S&P 500 (Q1 2025) | | | | | | | | |
| **Method** | **Nasdaq-100** | | | | **S&P 500** | | | |
| | RIC | RICIR | IC | ICIR | RIC | RICIR | IC | ICIR |
| Proxy Indicator (Diether et al., 2002) | 1.98 | 17.26 | 1.47 | 14.83 | 2.06 | 16.39 | 2.34 | 15.81 |
| LightGBM (Ke et al., 2017) | 2.40 | 18.75 | 2.38 | 19.36 | 2.64 | 19.42 | 2.47 | 17.38 |
| FactorVAE (Duan et al., 2022) | 3.42 | 27.86 | 3.29 | 27.05 | 3.55 | 24.60 | 3.49 | 27.85 |
| HireVAE (Wei et al., 2023) | 3.58 | 24.97 | 3.63 | 26.37 | 3.67 | 24.54 | 3.72 | 27.63 |
| DTML (Yoo et al., 2021) | 3.21 | 23.59 | 2.93 | 21.40 | 3.37 | 22.35 | 3.26 | 21.84 |
| MASTER (Li et al., 2024b) | 3.52 | 25.98 | 3.20 | 25.84 | 3.61 | 26.54 | 3.48 | 25.70 |
| SEP (Koa et al., 2024) | 3.43 | 26.35 | 3.19 | 25.76 | 0.62 | 6.35 | 0.74 | 5.89 |
| FinCON (Yu et al., 2024) | 3.48 | 25.82 | 3.63 | 25.97 | 1.13 | 8.56 | 0.97 | 6.75 |
| TradingAgents (Xiao et al., 2025b) | 3.50 | 26.76 | 3.71 | 26.99 | 3.78 | 28.04 | 3.92 | 29.31 |
| **MASS** | **3.96** | **29.84** | **4.01** | **27.53** | **4.05** | **29.73** | **3.99** | **29.67** |

2. **Maximum Drawdown (MDD)**
   The largest peak-to-trough decline in portfolio value, expressed as a percentage. It measures the strategy's downside risk, defined as:

$$\text{MDD} = \max\left(1 - \frac{V_t}{V_{\text{peak}}}\right)$$

where $V_t$ is the portfolio value at time $t$, and $V_{\text{peak}}$ is the maximum value before $t$.

3. **Sharpe Ratio (Sharpe)**
   A measure of risk-adjusted return, calculated as the excess return over the risk-free rate divided by the strategy's volatility:

$$\text{SR} = \frac{R_{\text{strategy}} - R_f}{\sigma_{\text{strategy}}}$$

where $R_f$ is the risk-free rate and $\sigma_{\text{strategy}}$ denotes the standard deviation of strategy returns. A higher value indicates superior risk-adjusted performance.

Table 7 provides experiment results on MASS and all baselines, demonstrating MASS's enhanced performance and stability.

A.5.2 RESULTS ON THE US STOCK MARKET

To evaluate the generalizability of MASS, we test it on the Nasdaq 100 and S&P 500 indices, using the same period as in Table 1. Table 8 shows that MASS performs effectively across these different asset classes.

## A.6 BACKTESTING EXPERIMENTS DETAILS

We conduct backtesting experiments on a simulated system. We conduct backtesting using a traditional index-enhancement strategy. The portfolio is rebalanced weekly, with a round-trip transaction cost of $0.1\%$. During the first fifteen minutes after the market opens on the first trading day of each week, we first exclude stocks that are limit-up or limit-down. Subsequently, we rank the portfolio construction signals and equally weight the top 20% of the ranked stocks. Stocks currently held but no longer in the top 20% are sold.

