# OpenReview forum: "MASS:Multi-Agent Simulation Scaling  for Portfolio Construction"
_ICLR.cc/2026/Conference — Submitted to ICLR 2026_

### Official Review · Reviewer_ym2m · 2025-10-15

**Soundness:** 2
**Presentation:** 2
**Contribution:** 2
**Rating:** 4
**Confidence:** 4

**Summary:**

The author introduces a framework, MASS, to construct optimal stock trading strategies using LLM based models in a multi-agent manner. A voting mechanism is leveraged to aggregate the strategies from multiple trading agents. Extensive evaluation is conducted with comprehensive selection of baselines. The empirical results, with varied performance metrics, shows that the proposed MASS framework consistently outperformed all the solid baselines.

**Strengths:**

- It is the first work that explores the influence of scales when leveraging multi-agent settings for stock trading.
- The evaluation is solid, with varied SoTA baseline methods, and all the claims are justified through solid testing results.
- Ablation study is conducted to sufficiently justify the choice of each component of the pipeline.
- The design of the entire framework is vivid, where varied agents are designed to represent individuals with different trading styles.

**Weaknesses:**

- The testing performance of the baseline methods at the beginning of 2025 is missing. Having such a comparison is important to justify the consistency of leading performance of MASS under the data-leakage-concern experiment.
- Though the overall experimental design is soundable and vivid, the contribution on the methodology aspect is slightly lacking, where the entire framework is heavily dominated by prompt engineering, and the proposed aggregation module is essentially a voting mechanism where the voting weight is adjusted based on the past performance.
- The complexity of the entire system is a concern for the feasibility of real-world deployment.

**Questions:**

- 1) Since the presentation is mostly emphasized on the performance, I'm wondering if there are any analysis or observation from the trading strategy perspective. For example, if running the multi-agents inference with multiple trails, will the similar strategy always retained (both for the aggregated strategy or each agent with varied trading style individually)?
- 2) Continued from the 1st question, if, hypothetically, the strategy will varied due to stochastic process in the LLM generation, to what extent it might influence the pattern of the optimal agent distribution identified by the backward optimization?
- 3) A final thought: since in the real market, if a strategy is known by more people, the expected return of such a strategy will reduce. Though this is slightly paradox with the disagreement hypothesis, is there any potential observation in the multi-agent environment, where there might be more diversity in the trading strategy when the overall return is relatively higher? Having these analysis will provide more insights on the behavior of the proposed framework.

---

> ### Author Response · Authors · 2025-11-17
>
> We sincerely thank you for your review and for recognizing the novelty of our work in exploring scaling effects, the solidity of our evaluation, and the vividness of the framework's design. Your insightful questions and feedback are invaluable in helping us improve the paper. Below, we address each of your concerns in detail.

---

> ### Author Response · Authors · 2025-11-17
>
> >**Weakness 1: The testing performance of the baseline methods at the beginning of 2025 is missing. Having such a comparison is important to justify the consistency of leading performance of MASS under the data-leakage-concern experiment.**
>
>
>
> Thank you for pointing this out. We agree that providing baseline performance on the 2025 dataset is essential to validate the consistency of MASS's leading performance and strengthen our data leakage analysis.
>
> In response, we have conducted extensive new experiments. We ran all baseline models on the same Q1 2025 dataset across all three Chinese A-share indices (SSE 50, CSI 300, and the new CSI A500). Furthermore, to demonstrate the broad generalizability of our findings, we also extended this analysis to the US market (Nasdaq 100 and S&P 500) for the same Q1 2025 period.
>
> The results, now included in the updated Table 1 in the main paper and Table 8 in  Appendix 5.2, are summarized below.
>
> MASS and all baselines on the Chinese stock market in 2025Q1:
>
> | Method              | SSE50    |           |          |           | CSI 300  |           |          |           | CSI A500 |           |          |           |
> | ------------------- | -------- | --------- | -------- | --------- | -------- | --------- | -------- | --------- | -------- | --------- | -------- | --------- |
> |                     | **RIC**  | **RICIR** | **IC**   | **ICIR**  | **RIC**  | **RICIR** | **IC**   | **ICIR**  | **RIC**  | **RICIR** | **IC**   | **ICIR**  |
> | **Proxy Indicator** | 1.46     | 10.60     | 1.51     | 9.89      | 1.52     | 10.37     | 2.01     | 14.28     | 1.04     | 9.75      | 0.98     | 9.97      |
> | **LightGBM**        | 1.66     | 12.35     | 1.58     | 11.73     | 1.59     | 8.79      | 1.85     | 11.97     | 1.77     | 12.84     | 1.58     | 12.60     |
> | **FactorVAE**       | 3.59     | 21.61     | 5.41     | 31.14     | 3.37     | 29.67     | 2.65     | 26.96     | 4.32     | 40.87     | 4.01     | 36.70     |
> | **HireVAE**         | 3.68     | 21.52     | 5.44     | 30.15     | 3.47     | 31.58     | 2.61     | 27.93     | 4.24     | 42.69     | 3.91     | 36.94     |
> | **DTML**            | 3.53     | 20.94     | 5.28     | 28.77     | 3.39     | 28.86     | 2.54     | 27.78     | 4.06     | 41.80     | 3.75     | 35.22     |
> | **MASTER**          | 3.70     | 21.38     | 5.49     | 30.26     | 3.46     | 29.74     | 2.58     | 28.47     | 4.13     | 45.52     | 3.89     | 36.67     |
> | **SEP**             | 3.65     | 20.92     | 5.47     | 29.99     | 1.45     | 10.06     | 0.84     | 9.76      | 4.25     | 46.31     | 3.96     | 38.75     |
> | **FinCON**          | 3.97     | 22.03     | 5.68     | 31.42     | 1.54     | 13.98     | 0.80     | 10.72     | 4.81     | 48.25     | 4.34     | 43.96     |
> | **TradingAgents**   | 4.02     | 21.94     | 5.71     | 31.99     | 3.63     | 29.80     | 2.97     | 30.63     | 4.86     | 48.95     | 4.20     | 43.94     |
> | **MASS**            | **4.50** | **24.41** | **6.12** | **38.33** | **3.91** | **37.44** | **3.36** | **34.56** | **5.19** | **56.17** | **4.66** | **48.82** |
>
> MASS and all baselines on the US stock market in 2025Q1:
>
> | Method              | Nasdaq-100 |           |          |           | S&P 500  |           |          |           |
> | ------------------- | ---------- | --------- | -------- | --------- | -------- | --------- | -------- | --------- |
> |                     | **RIC**    | **RICIR** | **IC**   | **ICIR**  | **RIC**  | **RICIR** | **IC**   | **ICIR**  |
> | **Proxy Indicator** | 1.98       | 17.26     | 1.47     | 14.83     | 2.06     | 16.39     | 2.34     | 15.81     |
> | **LightGBM**        | 2.40       | 18.75     | 2.38     | 19.36     | 2.64     | 19.42     | 2.47     | 17.38     |
> | **FactorVAE**       | 3.42       | 27.86     | 3.29     | 27.05     | 3.55     | 24.60     | 3.49     | 27.85     |
> | **HireVAE**         | 3.58       | 24.97     | 3.63     | 26.37     | 3.67     | 24.54     | 3.72     | 27.63     |
> | **DTML**            | 3.21       | 23.59     | 2.93     | 21.40     | 3.37     | 22.35     | 3.26     | 21.84     |
> | **MASTER**          | 3.52       | 25.98     | 3.20     | 25.84     | 3.61     | 26.54     | 3.48     | 25.70     |
> | **SEP**             | 3.43       | 26.35     | 3.19     | 25.76     | 0.62     | 6.35      | 0.74     | 5.89      |
> | **FinCON**          | 3.48       | 25.82     | 3.63     | 25.97     | 1.13     | 8.56      | 0.97     | 6.75      |
> | **TradingAgents**   | 3.50       | 26.76     | 3.71     | 26.99     | 3.78     | 28.04     | 3.92     | 29.31     |
> | **MASS**            | **3.96**   | **29.84** | **4.01** | **27.53** | **4.05** | **29.73** | **3.99** | **29.67** |
>
> As the results show, MASS gains enhanced performance on this unseen future data, across different markets. This provides evidence that its success is due to its adaptive framework, not data memorization.

---

> ### Author Response · Authors · 2025-11-17
>
> >**Weakness 2: Though the overall experimental design is soundable and vivid, the contribution on the methodology aspect is slightly lacking, where the entire framework is heavily dominated by prompt engineering, and the proposed aggregation module is essentially a voting mechanism where the voting weight is adjusted based on the past performance.**
>
> We appreciate you acknowledging the novelty of our experimental design. We would like to take this opportunity to elaborate on the core methodological contributions of MASS, which we believe are substantial and go far beyond prompt engineering or a simple voting mechanism.
>
> First, regarding the concern about prompt engineering, we want to clarify that MASS is architecturally designed to be robust and not dependent on fragile, finely-tuned prompts. We acknowledge that many recent agent-based systems (e.g., SEP, FinCON, TradingAgents) have made significant strides, often leveraging sophisticated prompting in conjunction with other powerful techniques—such as **reflection mechanisms or unique agent architectures**—to generate valuable insights.  Our framework offers a complementary path to robustness, where the strength stems primarily from its systemic design rather than from linguistic nuances. This is achieved in two key ways:
>
> 1. **Programmatic and Diversified Agent Generation:** We do not hand-craft unique prompts for each of our 512 agent instances. Instead, the system's heterogeneity is generated programmatically. As detailed in Section 3.1.1, we define a set of investment archetypes (e.g., risk appetite, holding period), and agent types are created from combinations of these characteristics. Furthermore, each agent is assigned a random, limited subset of stocks to analyze. This partitioned view of the market ensures a wide diversity of opinions and prevents the system's success from depending on a single, perfectly crafted "global" prompt. We use a generalized template that simply provides the necessary context: the agent's persona, its available information, and its objective.
> 2. **Empirical Validation of Prompt Insensitivity:** To empirically prove that our framework is not just a product of prompt engineering, we conducted an additional experiment. we took the prompts used for agent initialization and radically rephrased them, altering the wording, sentence structure, and tone, while keeping the core instructions the same.

---

> ### Author Response · Authors · 2025-11-17
>
> The detailed agent-initiated prompts before and after rephrasing are listed  below:
>
> ```text
> # original prompt
>
> Give the following input data:
> Input time-series data column name and their descriptions in JSON format(textual data example).
> latest macroeconomic and market insights.
> Please try to analyze and summarize an abstract investing style description.
> The output format is a json. The specific format of the output JSON is:
> {
>     "Outline": "The outline and general description for investment style within 50 words. The outline is a summarization about your investing strategy and your insights into the subsequent trend of the stock market, without any details below.",
>     "Details": {
>         "Risk Appetite": "conservative | moderate | moderately conservative | moderately aggressive | aggressive",
>         "Holding Period": "one day | about one week | about one month | about half a year | more than one year",
>
>         "Strategy Consistency": [0, 1] (Refers to the investor's ability to adhere to and execute their investment strategy with persistence and coherence, regardless of short-term market fluctuations or emotional influences. Higher number means high consistency",
>
>         "Rationality": [0, 1] (Refers to whether the investor's decision-making process is based on logic, data, and long-term objectives rather than emotions, biases, or short-term market noise. Higher number means high rationality",
>
>         "StockPoolSelector": "Specify what kind of preference you'd like to construct your watchlist stocks. The possible preferences are:
>
>   1. RandomStockSelector: Randomly construct your watchlist.
>
>   2. IndustryEqualStockSelector: Construct a stock pool with balanced distribution across industries.
>
>   3. MVEqualStockSelector: Construct a stock pool with balanced distribution across market capitalizations.
>
>   4.IndustryBasisStockSelector: Prefer stocks from specific industries and output the preferred industries. The result is presented in a list format.",
>
>         "Others": "Extra information about your investing strategy, maybe correlated with latest market and macroeconmic information and others. No more than 30 words."
>     }
> }
>
> {examples}
>
> Input data:
> {input_data}
>
> # changed prompt
>
> Task: Generate Investment Profile JSON
> OBJECTIVE: Synthesize a structured investment profile by analyzing provided data schema and market intelligence.
> INPUT PAYLOAD:
> data_schema: A JSON object defining available time-series data, including column names and descriptions.
> market_intelligence: A textual summary of the latest macroeconomic and market insights.
> CONSTRAINTS:
> The output MUST be a single, valid JSON object that strictly adheres to the following structure and constraints. No additional text or explanation is permitted outside the JSON structure.
>
> {
>     "Outline": "string // Max 50 words. A high-level summary of the investment strategy and market outlook. Must not contain details found in the 'Details' object.",
>     "Details": {
>         "Risk Appetite": "string // ENUM: Must be one of 'conservative', 'moderate', 'moderately conservative', 'moderately aggressive', 'aggressive'.",
>         "Holding Period": "string // ENUM: Must be one of 'one day', 'about one week', 'about one month', 'about half a year', 'more than one year'.",
>         "Strategy Consistency": "float // RANGE: [0.0, 1.0]. Represents adherence to strategy despite market noise. 1.0 is maximum consistency.",
>         "Rationality": "float // RANGE: [0.0, 1.0]. Represents logic-driven vs. emotional decision-making. 1.0 is maximum rationality.",
>         "StockPoolSelector": "string // ENUM: Must be one of 'RandomStockSelector', 'IndustryEqualStockSelector', 'MVEqualStockSelector', 'IndustryBasisStockSelector'. If 'IndustryBasisStockSelector' is chosen, specify preferred industries in a list.",
>         "Others": "string // Max 30 words. Supplementary notes on strategy, potentially linking to the micro-economic information and your insights into strategies."
>     }
> }
>
> {examples}
>
> Input data:
> {input_data}
> ```
>
>  We then re-ran the entire simulation on the SSE 50 index. The performance remained stable, as shown in the table below.
>
>
> | Prompt Version       | RIC  | RICIR | IC   | ICIR  |
> | :------------------- | :--- | :---- | :--- | :---- |
> | **Original Prompt**  | 8.16 | 41.74 | 5.90 | 33.43 |
> | **Rephrased Prompt** | 8.09 | 41.75 | 5.94 | 34.13 |
>
> This result indicates that the framework's success is not an artifact of specific linguistic phrasing. Instead, it underscores our central thesis: the power of MASS lies in the emergent intelligence derived from aggregating diverse perspectives and, most importantly, in the adaptive learning process that follows.

---

> ### Author Response · Authors · 2025-11-17
>
> Second, regarding the aggregation module, we agree that it can be viewed as a weighted voting system. However, we believe the true novelty lies not in the voting itself, but in a far more critical component: how the weights are dynamically learned. Unlike static systems or predefined workflows, MASS introduces a data-driven backward optimization process.
>
> This mechanism is the core engine of our framework. At the end of each trading day, it looks back at a recent window of market history. It then solves an optimization problem (Equation 5) to find the agent type distribution $ d_j $ that would have produced a portfolio signal most highly correlated with the actual market returns observed during that window. This is not just about rewarding past performance; it is a sophisticated process that learns the optimal blend of strategies that is best suited for the current market regime.
>
> It is this end-to-end, adaptive learning loop that is our primary methodological contribution. It transforms what would be a simple simulation into an intelligent, self-optimizing decision-making system that learns and adapts to market shifts in real-time. This dynamic, data-driven adaptation is a feature absent in prior work and is the key reason MASS can consistently generate excess returns, as visually demonstrated by the shifting agent distributions in Figure 5 during major market transitions.
>
> In summary, the key innovation of MASS is the synergy between large-scale multi-agent simulation and a dynamic backward optimization process. The simulation generates a rich diversity of opinions, and the optimization intelligently learns how to aggregate that diversity to achieve a specific goal. This creates an adaptive system that is more powerful and robust than the sum of its parts.

---

> ### Author Response · Authors · 2025-11-17
>
> > **Weakness 3: The complexity of the entire system is a concern for the feasibility of real-world deployment.**
>
> Thank you for raising this point about the system's complexity and its practical feasibility for real-world deployment. This is a vital concern for any proposed framework, and we designed MASS with this challenge explicitly in mind. We would like to offer a more detailed explanation of why MASS is both accessible and highly feasible.
>
> First, to address **setup complexity and promote transparency**, we have fully open-sourced the entire framework. This includes:
>
> - The complete source code for the MASS simulation and optimization.[`https://anonymous.4open.science/r/MASS-AC96`]
> - The comprehensive dataset for the SSE 50 index, including all firm-level features and macroeconomic data.[`https://anonymous.4open.science/r/MASS-AC96/stock_disagreement/dataset`]
> - The daily agent distribution snapshots from our experiments, allowing for full reproducibility of our results.[`https://anonymous.4open.science/r/MASS-AC96/ih_dist`]
>
> By making all components publicly available, we aim to demonstrate that the system is not an unmanageable black box. Researchers and practitioners can directly inspect, run, and verify our work, significantly lowering the barrier to entry for deployment.
>
> Second, to address **computational feasibility (both time and cost)**, it is essential to clarify the operational context of MASS.
>
> - **Online Learning at a Daily Frequency:** MASS is designed as an online learning system for daily portfolio rebalancing (a T+1 strategy), not high-frequency trading. All computations, including the forward propagation and backward optimization, occur *after* the market closes each day. This provides a generous window of over 12 hours for the daily update, which is more than sufficient.
> - **Efficient Optimization:** The backward optimization process might seem complex, but it is highly efficient in practice. We are not training a massive, high-dimensional model every day. Instead, the optimizer's sole task is to find the optimal weights for a single, low-dimensional vector (a 16-dimensional vector in our experiments) representing the agent type distribution. Because the search space is small, simulated annealing converges to a strong solution very quickly.
> - **Concrete Time and Cost Metrics:** In our implementation, the entire daily update cycle is remarkably fast and cheap.
>   - **Time:** A full daily update (one forward pass and one backward optimization with 100 iterations) takes **less than 10 minutes** to complete. This is a fraction of the 12+ hour window available, demonstrating that the system is far from being a computational bottleneck.
>   - **Monetary Cost:** The daily operational cost is minimal. As detailed in Appendix A.3.7, based on the Qwen2.5-72B API pricing on the Alibaba Cloud platform, the daily cost for running the simulation with 512 agents is approximately **$0.68 for the SSE 50**, **$2.27 for the CSI 300**, and **\$1.19 for the ChiNext 100**.
> - **Manageable Scalability:** The daily update cost has a linear complexity of $O(n_{type} \times n_{inv})$. This means the cost scales predictably and manageably with the number of agents. Even when scaling to 1024 agents, the daily API costs remain negligible compared to the potential excess returns generated by the system, as shown in our backtesting experiments.
>
> We have carefully engineered MASS to be practical. Its complexity is manageable due to our open-source release. Its computational demands are low, with daily updates completing in minutes for just a few dollars. This favorable cost-benefit profile makes MASS a highly feasible and attractive solution for real-world deployment by both independent researchers and financial institutions.
>
> We hope this detailed breakdown could alleviate your concerns about the system's practical feasibility.

---

> ### Author Response · Authors · 2025-11-17
>
> >**Question 1: Since the presentation is mostly emphasized on the performance, I'm wondering if there are any analysis or observation from the trading strategy perspective. For example, if running the multi-agents inference with multiple trails, will the similar strategy always retained (both for the aggregated strategy or each agent with varied trading style individually)?**
> >
> >**Question 2: Continued from the 1st question, if, hypothetically, the strategy will varied due to stochastic process in the LLM generation, to what extent it might influence the pattern of the optimal agent distribution identified by the backward optimization?**
>
> Thank you for this insightful question. You have raised a crucial point regarding the strategic consistency of MASS, and we appreciate the opportunity to provide a more detailed analysis from a trading strategy perspective.
>
> We agree that this is an important aspect to investigate. To address this, we have introduced a new case study in the revised paper (Appendix 3.5) that empirically investigates the stability of the aggregated strategy across multiple independent inference runs.
>
> To directly and comprehensively answer your question—"if running the multi-agents inference with multiple trails, will the similar strategy always retained"—we designed the following experiment:
>
> First, to quantify the strategy's focus on any given day, we define "stock popularity" as the weighted sum of agents selecting a stock, using the optimized agent distribution as weights (as described in Equation 3a). We then normalize these popularity scores across the stock universe for each day to create a daily popularity probability distribution. This distribution represents the aggregated strategic focus of the entire agent population.
>
> Next, we conducted two fully independent inference runs of MASS on the SSE 50 index for the entire 2023 period. The resulting popularity distributions from both runs are visualized in Figures 6a and 6b. As these figures show, we observe no significant divergence in the overall strategic patterns between the two runs.
>
> To enable a more granular analysis, we selected a representative subset of nine stocks spanning diverse styles (value, growth, and beta) provided by the [cs index](https://www.csindex.com.cn/#/), as detailed below and Table 3 in the Appendix . Their popularity evolutions are magnified and presented in Figures 6c and 6d. From these, we can draw two key conclusions:
>
>
> | Style      | Ticker | Company                                           |
> | :--------- | :----- | :------------------------------------------------ |
> | **Value**  | 600036 | China Merchants Bank Co., Ltd.                    |
> | **Value**  | 601857 | PetroChina Company Limited                        |
> | **Value**  | 601088 | China Shenhua Energy Company Limited              |
> | **Growth** | 600519 | Kweichow Moutai Co., Ltd.                         |
> | **Growth** | 600276 | Jiangsu Hengrui Pharmaceuticals Co., Ltd.         |
> | **Growth** | 600309 | Wanhua Chemical Group Co., Ltd.                   |
> | **Beta**   | 601888 | China Tourism Group Duty Free Corporation Limited |
> | **Beta**   | 603288 | Foshan Haitian Flavouring & Food Co., Ltd.        |
> | **Beta**   | 603259 | WuXi AppTec Co., Ltd.                             |
>
> 1. We do observe subtle variations between the two runs, which can be attributed to the inherent stochasticity of LLM-based generation at the individual agent level. However, we posit that these minor individual variations are  smoothed out at the aggregate level due to two primary factors: (1) The large number of agents in our simulation provides a diversifying effect that mitigates randomness. (2) More importantly, our backward optimization mechanism plays a crucial role in steering the overall agent distribution, ensuring that the emergent, aggregated strategy follows a consistent paradigm.
>
> 2. This consistency is evident in the evolutionary patterns of the nine stocks, which are broadly similar across both runs. For instance, in both Figures 6c and 6d, the popularity of value-style stocks shows a distinct upward trend throughout 2023. We cautiously suggest that our backward optimization mechanism successfully identified the shift in market preference towards value during this period, which is a potential key reason for MASS's ability to outperform the market.
>
>    Furthermore, while the high-level, aggregated stock selections and investment styles of MASS remain consistent, we observe that sampling from multiple runs of a single agent reveals variance in its stock choices. This can occasionally lead to slight style drift or selections that diverge from the model's overarching macro-outlook. This behavior occurs because our optimization method selects agents based solely on their performance within the most recent window.
>
> We hope this detailed analysis and the new case study effectively address your question about the strategic consistency of our framework.

---

> ### Author Response · Authors · 2025-11-17
>
> >**Question 3: A final thought: since in the real market, if a strategy is known by more people, the expected return of such a strategy will reduce. Though this is slightly paradox with the disagreement hypothesis, is there any potential observation in the multi-agent environment, where there might be more diversity in the trading strategy when the overall return is relatively higher? Having these analysis will provide more insights on the behavior of the proposed framework.**
>
>
>
> This is a fascinating and profound question that gets to the heart of the efficient market hypothesis and the challenge of alpha decay. Thank you for this thought-provoking point. You are right to question how a system that rewards consensus can avoid the pitfalls of a popular strategy.
>
> To address the paradox you astutely identified, it is crucial to distinguish between two concepts: **1) External strategy popularity**, which causes alpha decay in the real market, and **2) Internal agent consensus**, which MASS uses as a proprietary signal.
>
> 1. **Internal Consensus is a Signal of Conviction, Not a Crowded Trade:** The "alpha decay" you describe happens when a large number of independent, external market participants discover and execute the same strategy, causing the opportunity to be arbitraged away. The consensus in MASS operates on a completely different level. Our agents are not the market; they are internal components of a single, unified meta-strategy.
> 2. Think of MASS not as a single strategy, but as a sophisticated hedge fund's investment committee. The committee is composed of diverse analysts (our agents), each with a unique philosophy. When different analysts, using their distinct models and data, independently arrive at the same stock pick, the committee's confidence (our `Signal`) increases. This internal consensus is a private, proprietary signal of robustness, not a public, crowded trade. It suggests the investment thesis is sound from multiple perspectives, making it *more likely* to generate future returns, not less.
> 3. **Dynamic Diversity as a Defense Against Alpha Decay:** This leads directly to your question about the relationship between strategy diversity and returns. Our framework suggests the relationship is not static ("more diversity is always better") but dynamic, and it is governed by the backward optimization process. This is our core defense against strategy decay.
>    1. **During Stable Regimes (Potentially Higher Returns):** In periods of high, stable returns where a particular market style is clearly dominant, MASS may actually *decrease* its effective diversity. It does this by concentrating its weights on the few agent types that are best aligned with the prevailing market regime. It is exploiting the current source of alpha.
>    2. **During Market Shifts (Uncertainty):** Conversely, during market shifts or periods of high uncertainty, as one strategy begins to fail, its performance in the lookback window will suffer. The backward optimization process will then naturally *increase* diversity by reducing weight on the failing strategy and spreading it more evenly across other agent types. In this state, the system is actively "exploring" to find the new winning strategy for the next regime.
> 4. This dynamic is precisely what is visualized in Figure 5. The rapid shifts in agent distribution at points A and B represent periods of high strategic exploration. In the more stable periods between these shifts, the distribution is less varied as the system converges on the optimal strategy mix for that environment.
>
> In essence, MASS doesn't just *have* diversity; it **manages** it as a risk and exploration tool. It uses internal consensus as a conviction signal while using its adaptive diversity to constantly search for new sources of alpha. This continuous cycle of exploiting current alpha and exploring for future alpha is precisely how the framework is designed to combat the very problem of strategy decay that you have highlighted.

---

> ### Author Response · Authors · 2025-11-17
>
> Thank you again for your invaluable guidance. We have incorporated new experiments and clarifications in an effort to fully address the points you raised. If you have any further questions, please feel free to contact us. We hope you will find the revised paper much improved and would be deeply grateful for your reconsideration.

---

> > ### Comment · Reviewer_ym2m · 2025-11-19
> > **Response to Authors**
> >
> > I thank the authors for the comprehensive response. All of my concerns are addressed adequately through extensive experimental results, which includes:
> > - Presenting the performance comparison at the beginning of 2025.
> > - Ablation study justifying the extent of influence on performance from varied prompts.
> > - Case study exploring at the aspect of strategy and the stability of the learned outcome.
> >
> > With these additions, the revised manuscript is much more well-rounded. From the general response, I also learned that further validations were conducted on several additional stocks, which provides stronger evidence for the generalizability of the approach.
> >
> > Overall, the work now presents clear motivation, clearly described methodology, in-depth experimentation under varied settings with thorough ablations, and detailed case studies with insightful discussion. I recommend accepting this paper, which is reflected in my updated rating. Thank you again for your solid study.

---

> > > ### Author Response · Authors · 2025-11-19
> > >
> > > Dear Reviewer ym2m,
> > >
> > > Thank you so much for your positive feedback and for raising the score. Your recognition truly made my day!
> > >
> > > Wish you all the best!
> > >
> > > Sincerely, Authors

---

### Official Review · Reviewer_xnHx · 2025-10-28

**Soundness:** 2
**Presentation:** 2
**Contribution:** 2
**Rating:** 4
**Confidence:** 5

**Summary:**

This paper proposes MASS, a multi-agent simulation framework that directly constructs investment portfolios without predicting stock prices or relying on static decision workflows. It adaptively adjusts the distribution of LLM-based agents via backward optimization, enabling responsiveness to changing market regimes. Experiments on the 2023 China A-share market show that MASS scales effectively with more agents and outperforms multiple SOTA baselines in profitability and robustness.

**Strengths:**

1. The idea of modeling investors as heterogeneous LLM-based agents with diverse preferences and strategies is conceptually appealing.
2. The release of code and dataset enhances openness and encourages further research in this emerging area.

**Weaknesses:**

1. The use of simulated annealing as an optimizer is conceptually inconsistent with a dynamic market learning framework. SA requires hundreds of perturbations for convergence, whereas financial markets demand fast, step-wise adaptation. This slow, global optimization process fundamentally conflicts with the model’s claim of online learning and timely market responsiveness. Moreover, in MASS’s experiments, the dataset only covers stock market daily data from 2023, providing merely a few hundred data steps. Given such a limited time horizon, can SA truly converge in practice?
2. Market disagreement hypothesis plays a critical underlying role in the MASS portfolio management method. However, the definition of the signal in Eq4 implies that higher disagreement reduces the resulting signal, leading to a negative portfolio allocation for the stock when applying the top-K strategy. This is somehow inconsistent with the market disagreement hypothesis, which states that greater disagreement may actually lead to higher prices due to optimistic investors dominating valuation under short-selling constraints.

3. MASS evaluates performance using IC, RankIC, and related correlation metrics to measure alignment between its signals and realized returns. This experimental design is essentially identical to that used in classic factor-based models [1, 2], where factors are constructed to predict future returns and assessed through the same correlation metrics.

However, the paper explicitly states that the framework is not intended for price prediction (lines 90–92). Given this positioning, it would be beneficial to clarify the relationship between MASS and traditional factor-mining approaches. In particular, including representative factor model methods [1, 2] as baselines could help demonstrate whether the proposed multi-agent simulation captures information beyond conventional return-predictive factors. Otherwise, it remains somewhat unclear how much of the reported performance improvement should be attributed to the agent-based design.

​
[1] [22'AAAI]FactorVAE: A Probabilistic Dynamic Factor Model Based on Variational Autoencoder for Predicting Cross-Sectional Stock Returns
[2] [23'IJCAI]HireVAE: An Online and Adaptive Factor Model Based on Hierarchical and Regime-Switch VAE
some concerns addressed in weakness part

**Questions:**

Some concerns addressed in weakness part

---

> ### Author Response · Authors · 2025-11-17
>
> Thank you for your valuable feedback. We are pleased that you found our LLM-based agent approach to be "conceptually appealing" and appreciated our efforts to encourage further research by releasing our code and dataset. We will now address your comments point by point.

---

> ### Author Response · Authors · 2025-11-17
>
> > **Weakness 1: The use of simulated annealing as an optimizer is conceptually inconsistent with a dynamic market learning framework. SA requires hundreds of perturbations for convergence, whereas financial markets demand fast, step-wise adaptation. This slow, global optimization process fundamentally conflicts with the model’s claim of online learning and timely market responsiveness. Moreover, in MASS’s experiments, the dataset only covers stock market daily data from 2023, providing merely a few hundred data steps. Given such a limited time horizon, can SA truly converge in practice?**
>
> Thank you for raising this point about the practical feasibility of our optimization process. We would like to clarify our framework's design and operational context:
>
> 1. **Online Learning at a Daily Frequency:** MASS is designed for daily portfolio rebalancing (a T+1 strategy), not high-frequency trading. The optimization process occurs after the market closes each day to determine the agent distribution for the next trading day. This provides a window of over 12 hours for computation. As detailed in Appendix Table 6 of our submission, the entire daily update process (forward pass and backward optimization) takes less than 10 minutes. This is fully consistent with an "online learning" paradigm at a daily frequency.
>
> 2. **Optimization Target and Convergence:** The backward optimization does not train a complex, high-dimensional model. Instead, it optimizes a single, low-dimensional vector representing the agent distribution (a 16-dimensional vector in our experiments). Because the optimization target is simple, simulated annealing can explore the solution space and converge to a strong solution relatively quickly. To be more specific, the simulated annealing process does not take a single, small step. Instead, it performs an iterative search within each daily optimization window. On any given day, the optimizer starts with an agent distribution and then generates and evaluates a series of new candidate distributions by repeatedly perturbing the current one. Each of these 'perturb-and-evaluate' cycles constitutes an 'iteration'. In our setup, this process is repeated 100 times. The final output of this extensive 100-iteration search is the *single*, optimized agent distribution that is carried forward to the next day. Therefore, a single daily update can reflect a significant change from the previous day's strategy, as it is the result of a thorough intra-day search, not a single incremental step.
>
>    To further address your question regarding convergence, we have conducted additional experiments on parameter sensitivity, specifically for the cooling rate and `max_iterations`. As shown in the table below (and detailed in Table 4 of the Appendix), our results for `max_iterations` indicate that the model achieves a promising and converged result as long as the number of iterations is 100 or greater.
>
>
> | Cooling rate |      RIC |     RICIR |       IC |      ICIR |
> | ------------ | -------: | --------: | -------: | --------: |
> | 1.00         |    -0.16 |     -3.58 |    -0.27 |     -4.99 |
> | 0.98         |     5.79 |     39.68 |     4.21 |     30.81 |
> | 0.95         |     6.50 |     43.49 |     4.65 |     33.32 |
> | 0.90         | **6.53** | **44.82** | **4.77** | **34.06** |
> | 0.85         |     5.81 |     40.12 |     4.16 |     31.13 |
> | 0.80         |     4.12 |     31.90 |     3.89 |     24.58 |
>
> | Max iteration |      RIC |     RICIR |       IC |      ICIR |
> | ------------- | -------: | --------: | -------: | --------: |
> | 0             |     0.36 |      5.36 |     0.41 |      6.69 |
> | 25            |     3.04 |     23.55 |     2.94 |     21.89 |
> | 50            |     4.69 |     31.80 |     3.73 |     26.66 |
> | 100           |     6.50 | **43.49** |     4.65 | **33.32** |
> | 200           | **6.53** |     42.76 | **4.66** |     32.91 |
>
> In summary, MASS operates within a practical daily timeframe, optimizes a low-dimensional target that allows for rapid convergence, and is robust to its optimization settings.
> We hope our clarification can answer your questions. If you have any further questions, feel free to contact us!

---

> ### Author Response · Authors · 2025-11-17
>
> >**Weakness 2: Market disagreement hypothesis plays a critical underlying role in the MASS portfolio management method. However, the definition of the signal in Eq4 implies that higher disagreement reduces the resulting signal, leading to a negative portfolio allocation for the stock when applying the top-K strategy. This is somehow inconsistent with the market disagreement hypothesis, which states that greater disagreement may actually lead to higher prices due to optimistic investors dominating valuation under short-selling constraints.**
>
>
>
> Thank you for this observation. This gives us an opportunity to clarify a subtle but crucial distinction in the interpretation of the Market Disagreement Hypothesis (MDH).
>
> You are correct that the MDH, as proposed by Miller (1977) [1]**,** suggests that under short-sale constraints, stocks with higher disagreement will have higher *current prices*. This is because only the opinions of optimistic investors are fully reflected in the price, leading to potential overvaluation.
>
> However, our goal is not to identify stocks with high current prices, but to identify stocks with high *future returns*. A stock that is currently overpriced due to disagreement-fueled optimism is, by definition, more likely to experience lower future returns as its price reverts to a more fundamental value.
>
> Therefore, our signal formulation in Equation 4 is perfectly consistent with this dynamic. By penalizing disagreement, we are actively filtering out these potentially overvalued stocks. We reward consensus to identify stocks that a diverse set of our optimized agent archetypes agree on, which are more likely to be fairly valued or undervalued and thus poised for higher future returns.
>
> To empirically validate this, we can refer to the ablation study in our original paper (**Table 3, "w/o MDH"**). Removing the disagreement term and relying only on consensus degrades performance, confirming that accounting for disagreement is critical. To further address your concern, we ran an additional experiment comparing our `Consensus - Disagreement` formulation against `Consensus Only` and `Consensus + Disagreement`. Our results on the SSE 50 and NASDAQ 100 datasets show that subtracting the disagreement term from the consensus consistently improves performance. Interestingly, we find the contribution of the disagreement term to be less pronounced on the NASDAQ 100. We attribute this to the mature financial infrastructure of the U.S. market, where a wide array of instruments—including short selling, leveraged ETFs, and various derivatives—provides bearish investors with ample avenues to express their views, thereby diminishing the relative impact of the disagreement captured by our signal.
>
>
> | Aggregation Method                  | SSE50 RIC | SSE50 RICIR | Nasdaq-100 RIC | Nasdaq-100 RICIR |
> | :---------------------------------- | :-------- | :---------- | :------------- | :--------------- |
> | **Consensus + Disagreement**        | 4.73      | 26.80       | 3.98           | 30.31            |
> | **Consensus Only**                  | 6.28      | 32.68       | 4.02           | 30.69            |
> | **Consensus - Disagreement (MASS)** | **8.16**  | **41.74**   | **4.27**       | **31.05**        |
>
> We hope that our explanation could help address your concerns. If you have any further questions, we would be more than happy to discuss them.
>
> References:
>
> [1] Edward M Miller. Risk, uncertainty, and divergence of opinion. The Journal of Finance, 32(4) 1977

---

> ### Author Response · Authors · 2025-11-17
>
> > **Weakness 3: MASS evaluates performance using IC, RankIC, and related correlation metrics to measure alignment between its signals and realized returns. This experimental design is essentially identical to that used in classic factor-based models [1, 2], where factors are constructed to predict future returns and assessed through the same correlation metrics.** **However, the paper explicitly states that the framework is not intended for price prediction (lines 90–92). Given this positioning, it would be beneficial to clarify the relationship between MASS and traditional factor-mining approaches. In particular, including representative factor model methods [1, 2] as baselines could help demonstrate whether the proposed multi-agent simulation captures information beyond conventional return-predictive factors. Otherwise, it remains somewhat unclear how much of the reported performance improvement should be attributed to the agent-based design.**
>
>
>
> Thank you for this excellent suggestion. We agree that comparing MASS to strong factor-model baselines is essential for contextualizing its performance and demonstrating its unique contribution.
>
> In response to your feedback, we have  expanded our experimental section in the revised paper.
>
> 1. **New Baselines Added:** We have added two state-of-the-art factor models, **FactorVAE** [1] and **HireVAE** [2], to our set of baselines (Table 1 \& Table 7). Besides, we added discussions about the two baselines in the related work (Section 2.1), and plotted the backtest curve in Section 4.1.3.
>
> 2. **Comparative Results:** As shown in the **updated Table 1 and below**, MASS consistently achieves enhanced performance, while FactorVAE and HireVAE also show promising results.
>
>    2.1 The A-share stock market in 2023
>
> | Stock pool       | model         | RIC      | RICIR     | IC       | ICIR      |
> | :--------------- | :------------ | :------- | :-------- | :------- | :-------- |
> | **SSE50**        | **FactorVAE** | 5.05     | 38.27     | 4.89     | 26.56     |
> |                  | **HireVAE**   | 5.17     | 29.06     | 5.02     | 29.93     |
> |                  | **MASS**      | **8.16** | **41.74** | **5.90** | **33.43** |
> | **CSI 300**      | **FactorVAE** | 4.95     | 34.89     | 4.16     | 31.13     |
> |                  | **HireVAE**   | 5.23     | 36.21     | 4.22     | 31.08     |
> |                  | **MASS**      | **6.50** | **43.49** | **4.65** | **33.32** |
> | **Chi Next 100** | **FactorVAE** | 3.98     | 28.69     | 4.03     | 29.35     |
> |                  | **HireVAE**   | 4.03     | 32.25     | 4.14     | 30.08     |
> |                  | **MASS**      | **7.62** | **62.87** | **6.28** | **55.88** |
>
> 2.2  The A-share stock market in Q1 2025
>
> | Stock pool   | Model         | RIC      | RICIR     | IC       | ICIR      |
> | :----------- | :------------ | :------- | :-------- | :------- | :-------- |
> | **SSE50**    | **FactorVAE** | 3.59     | 21.61     | 5.41     | 31.14     |
> |              | **HireVAE**   | 3.68     | 21.52     | 5.44     | 30.15     |
> |              | **MASS**      | **4.50** | **24.41** | **6.12** | **38.33** |
> | **CSI 300**  | **FactorVAE** | 3.37     | 29.67     | 2.65     | 26.96     |
> |              | **HireVAE**   | 3.47     | 31.58     | 2.61     | 27.93     |
> |              | **MASS**      | **3.91** | **37.44** | **3.36** | **34.56** |
> | **CSI A500** | **FactorVAE** | 4.32     | 40.87     | 4.01     | 36.70     |
> |              | **HireVAE**   | 4.24     | 42.69     | 3.91     | 36.94     |
> |              | **MASS**      | **5.19** | **56.17** | **4.66** | **48.82** |

---

> ### Author Response · Authors · 2025-11-17
>
> We have also included these new baselines in our **cross-market experiments on the US (Nasdaq 100, S&P 500)** and our **data leakage tests on Q1 2025 data** (see Appendix 5.2) listed below.
>
> 3.1 The US stock market in 2023
>
> | Stock pool     | Model         | RIC      | RICIR     | IC       | ICIR      |
> | :------------- | :------------ | :------- | :-------- | :------- | :-------- |
> | **Nasdaq-100** | **FactorVAE** | 3.49     | 26.05     | 3.62     | **28.95**     |
> |                | **HireVAE**   | 3.52     | 25.30     | 3.79     | 27.98     |
> |                | **MASS**      | **4.27** | **31.05** | **3.94** | 28.90 |
> | **S&P 500**    | **FactorVAE** | 3.96     | 28.34     | 3.77     | 29.64     |
> |                | **HireVAE**   | 4.12     | 27.86     | 3.83     | 28.39     |
> |                | **MASS**      | **4.31** | **31.45** | **3.95** | **28.68** |
>
> 3.2 The US stock market in Q1 2025
>
> | Stock pool     | Factor        | RIC      | RICIR     | IC       | ICIR      |
> | :------------- | :------------ | :------- | :-------- | :------- | :-------- |
> | **Nasdaq-100** | **FactorVAE** | 3.42     | 27.86     | 3.29     | 27.05     |
> |                | **HireVAE**   | 3.58     | 24.97     | 3.63     | 26.37     |
> |                | **MASS**      | **3.96** | **29.84** | **4.01** | **27.53** |
> | **S&P 500**    | **FactorVAE** | 3.55     | 24.60     | 3.49     | 27.85     |
> |                | **HireVAE**   | 3.67     | 24.54     | 3.72     | 27.63     |
> |                | **MASS**      | **4.05** | **29.73** | **3.99** | **29.67** |

---

> ### Author Response · Authors · 2025-11-17
>
> Thank you again for your thoughtful reviews and efforts to help us refine our manuscript. Your feedback is very important to us. If you have any other questions, we are always happy to assist you!

---

> ### Author Response · Authors · 2025-11-26
> **Humbly Requesting Your Feedback**
>
> Dear Reviewer xnHx:
>
> We deeply appreciate you sharing your valuable feedback and insights! We hope that the responses we provided have satisfactorily addressed all your concerns. As the Discussion Period will officially end on December 2nd (AoE), we just want to confirm that everything is fully resolved for you.
>
> We sincerely strive to earn your continued recognition for our efforts.
>
> Best Regards,
>
> The Authors

---

> > ### Author Response · Authors · 2025-11-28
> >
> > Dear Reviewer xnHx,
> >
> > Thank you again for your invaluable time and expertise.
> >
> > In direct response to your points, our summary of changes is as follows: (1)We added two SOTA factor models as baselines. (2)We also clarified the practical feasibility of simulated annealing in our daily online-learning setting and added a new sensitivity analysis to confirm its convergence. (3)We clarified the interpretation of the MDH and provided a new ablation study to validate our signal aggregation.
> >
> > These additions and clarifications have been incorporated into the revised manuscript.
> >
> > Your guidance has been invaluable in strengthening this work.  We hope our response and the revised paper could address your concerns. We would be grateful for your response and any further thoughts.
> >
> >
> > Respectfully,
> >
> > The Authors

---

### Official Review · Reviewer_EMCk · 2025-10-31

**Soundness:** 2
**Presentation:** 2
**Contribution:** 2
**Rating:** 4
**Confidence:** 3

**Summary:**

The paper introduces the Multi-Agent Scaling Simulation , a framework that employs multi-agent simulations for direct, end-to-end portfolio construction. Unlike traditional approaches that rely on intermediate steps like stock movement predictions or predefined workflows, MASS optimizes the distribution of heterogeneous agents in a backward optimization process, enabling adaptation to changing market conditions.

**Strengths:**

Originality: The introduction of a multi-agent simulation framework for direct portfolio construction is highly original. It moves beyond the typical stock prediction models by creating a dynamic system that adapts to market changes, offering a new approach to portfolio optimization.

**Weaknesses:**

Lack of Deep Contextualization: The paper does not fully contextualize how MASS compares to other multi-agent simulation approaches in finance, nor does it explore the potential limitations of the approach in different financial contexts (e.g., different asset classes or regimes).

Complexity of Agent Interactions: While the paper introduces a sophisticated framework, the interaction between agents and their impact on portfolio performance could be explained more clearly.

**Questions:**

Are there any limitations to the scaling effect, such as diminishing returns at very high numbers of agents? A more detailed analysis of potential trade-offs would be valuable.

---

> ### Author Response · Authors · 2025-11-17
>
> Thank you for your valuable review. We are delighted that you appreciated the originality of our work, particularly viewing our multi-agent framework as a new approach to direct portfolio construction. Your feedback has been very helpful, and we will now address your concerns step by step.

---

> ### Author Response · Authors · 2025-11-17
>
> > **Weakness 1: Lack of Deep Contextualization: The paper does not fully contextualize how MASS compares to other multi-agent simulation approaches in finance, nor does it explore the potential limitations of the approach in different financial contexts (e.g., different asset classes or regimes).**
>
>
>
> Thank you for this point. We recognize the need to better position MASS within the existing landscape and demonstrate its generalizability.
>
> **Contextualization against other Multi-Agent Simulations:** In Section 2.3 of our submission, we began to draw this distinction. We have now expanded this discussion to more explicitly highlight MASS's unique contributions. Unlike prior financial simulation works StockAgent [1], or TwinMarket[2], which primarily focus on simulating emergent market phenomena for analytical insight, MASS is an **application-oriented framework**. The key differentiator is our **backward optimization process**. Instead of observing emergent behavior, we use the simulation's aggregated output to make concrete investment decisions and then use market feedback to learn the optimal way to combine agent opinions. This shifts the paradigm from analyzing a simulation to optimizing a simulation for a real-world task. While Mars[3] is also a recent and relevant work in market simulation, our approaches differ in scope. Mars provides a micro-level simulation by modeling the order book and the dynamics of individual transactions. In contrast, our work takes a macro-level perspective, simulating the collective behavior of investors as market participants.
>
> We now provide a revised version in Section 2.3, offering a clearer comparison.
>
> **Generalizability across Different Financial Contexts:** We agree that demonstrating performance beyond a single market is crucial.
>
> - **Intra-Market Diversity:** As noted in the paper, our initial experiments already spanned three stylistically diverse Chinese indices: the SSE 50 (large-cap value), CSI 300 (broad market), and ChiNext 100 (growth/tech). The consistent outperformance of MASS across these different regimes provides initial evidence of its adaptability.
>
> - **Cross-Market Validation:** To directly address your concern about transferability, and in response to feedback from you and another reviewer, we have conducted **extensive new experiments on the US market**. We evaluated MASS against all baselines on the **Nasdaq 100** and **S&P 500** indices for the same 2023 and Q1 2025 periods. The data was sourced from public repositories (Microsoft Qlib, Yahoo Finance). Besides, we add two new SOTA baselines,  FactorVAE[4] and HireVAE[5]. The results, which have been added to Appendix 5.2 (Table 8) and are summarized below, show that MASS continues to significantly outperform the state-of-the-art baselines, confirming its robustness in a completely different market environment.
>
>   Main experiments on the US markets (2023)
>
> | Method                     | Nasdaq-100 RIC | Nasdaq-100 RICIR | Nasdaq-100 IC | Nasdaq-100 ICIR | S&P 500 RIC | S&P 500 RICIR | S&P 500 IC | S&P 500 ICIR |
> | -------------------------- | -------------: | ---------------: | ------------: | --------------: | ----------: | ------------: | ---------: | -----------: |
> | **Proxy Indicator (2002)** |           1.94 |            15.37 |          1.82 |           13.91 |        1.85 |         16.02 |       1.93 |        14.31 |
> | **LightGBM (2017)**        |           2.71 |            19.90 |          2.56 |           19.34 |        2.06 |         19.84 |       2.19 |        17.83 |
> | **FactorVAE (2022)**       |           3.49 |            26.05 |          3.62 |       **28.95** |        3.96 |         28.34 |       3.77 |        29.64 |
> | **HireVAE(2023)**          |           3.52 |            25.30 |          3.79 |           27.98 |        4.12 |         27.86 |       3.83 |        28.39 |
> | **DTML (2021)**            |           3.15 |            22.90 |          2.83 |           21.56 |        3.52 |         24.65 |       2.96 |        20.10 |
> | **MASTER (2024)**          |           3.38 |            23.62 |          2.98 |           21.49 |        3.27 |         25.93 |       3.09 |        22.53 |
> | **SEP (2024)**             |           3.40 |            22.99 |          3.26 |           23.85 |        1.38 |         11.82 |       0.82 |         7.81 |
> | **FinCON (2024)**          |           3.46 |            23.81 |          3.24 |           24.77 |        1.24 |         10.27 |       0.68 |         8.64 |
> | **TradingAgents (2025)**   |           3.63 |            27.36 |          3.85 |           28.29 |        4.07 |         31.28 |       3.89 |        27.94 |
> | **MASS**                   |       **4.27** |        **31.05** |      **3.94** |           28.90 |    **4.31** |     **31.45** |   **3.95** |    **28.68** |

---

> ### Author Response · Authors · 2025-11-17
>
> Data leakage experiments on US markets (Q1 2025)
>
> | Method                                     | Nasdaq-100 RIC | Nasdaq-100 RICIR | Nasdaq-100 IC | Nasdaq-100 ICIR | S&P 500 RIC | S&P 500 RICIR | S&P 500 IC | S&P 500 ICIR |
> | ------------------------------------------ | -------------: | ---------------: | ------------: | --------------: | ----------: | ------------: | ---------: | -----------: |
> | **Proxy Indicator (Diether et al., 2002)** |           1.98 |            17.26 |          1.47 |           14.83 |        2.06 |         16.39 |       2.34 |        15.81 |
> | **LightGBM (2017)**                        |           2.40 |            18.75 |          2.38 |           19.36 |        2.64 |         19.42 |       2.47 |        17.38 |
> | **FactorVAE (2022)**                       |           3.42 |            27.86 |          3.29 |           27.05 |        3.55 |         24.60 |       3.49 |        27.85 |
> | **HireVAE(2023)**                          |           3.58 |            24.97 |          3.63 |           26.37 |        3.67 |         24.54 |       3.72 |        27.63 |
> | **DTML (2021)**                            |           3.21 |            23.59 |          2.93 |           21.40 |        3.37 |         22.35 |       3.26 |        21.84 |
> | **MASTER (2024)**                          |           3.52 |            25.98 |          3.20 |           25.84 |        3.61 |         26.54 |       3.48 |        25.70 |
> | **SEP (2024)**                             |           3.43 |            26.35 |          3.19 |           25.76 |        0.62 |          6.35 |       0.74 |         5.89 |
> | **FinCON (2024)**                          |           3.48 |            25.82 |          3.63 |           25.97 |        1.13 |          8.56 |       0.97 |         6.75 |
> | **TradingAgents (2025)**                   |           3.50 |            26.76 |          3.71 |           26.99 |        3.78 |         28.04 |       3.92 |        29.31 |
> | **MASS**                                   |       **3.96** |        **29.84** |      **4.01** |       **27.53** |    **4.05** |     **29.73** |   **3.99** |    **29.67** |
>
>
>
> References:
>
> [1] Zhang, Chong, et al. "When ai meets finance (stockagent): Large language model-based stock trading in simulated real-world environments." *arXiv preprint arXiv:2407.18957* (2024).
>
> [2] Yang, Yuzhe, et al. "TwinMarket: A Scalable Behavioral and Social Simulation for Financial Markets." Neurips 2025.
>
> [3] Yuan, Huizhuo, Yifeng Liu, Shuang Wu, Xun Zhou, and Quanquan Gu. "Mars: Unleashing the power of variance reduction for training large models."  ICLR 2025.
>
> [4]Duan, Yitong, et al. "Factorvae: A probabilistic dynamic factor model based on variational autoencoder for predicting cross-sectional stock returns." AAAI 2022.
>
> [5] Wei, Zikai, et al. "Hirevae: An online and adaptive factor model based on hierarchical and regime-switch vae." IJCAI 2023.
>
>
>
> We hope that the expanded discussion and these new cross-market experiments can provide the compelling evidence of contextualization and generalizability that you suggested.

---

> ### Author Response · Authors · 2025-11-17
>
> >**Weakness 2:** Complexity of Agent Interactions: While the paper introduces a sophisticated framework, the interaction between agents and their impact on portfolio performance could be explained more clearly.
>
>
>
> Thank you for highlighting the need for more clarity on this mechanism. We would like to explain this at both a macro and a micro level.
>
> 1. **Macro-Level Interaction and Impact:** It is important to clarify that agents in MASS interact **indirectly**. There is no direct communication between them. Instead, their "interaction" occurs at the aggregation stage (Equation 3 & 4), where their individual decisions are synthesized into a collective market view. The impact of this interaction is then governed by the **backward optimization process**, which learns the optimal weight (distribution \( d \) ) for each agent type based on historical performance. Figure 5 in our paper provides a macro-level visualization of this process in action: it shows how the aggregate agent distribution dynamically shifts in response to major market regime changes, enabling the system to maintain stable excess returns.
>
> 2. **Micro-Level Case Study:** To provide a more granular view, and prompted by your valuable feedback, we have added a new case study in Appendix A.3.5. This case study zooms in on the market shift in August 2023 (marked 'B' in Figure 5) and analyzes the reasoning of two specific agent types:
>
>    1. **Agent O (Value/Dividend Focus):** Guided by its mandate, this agent interpreted macro data as favorable for dividend assets and selected stocks like China Shenhua Energy.
>
>    2. **Agent D (Consumption/Recovery Focus):** This agent's logic led it to anticipate a consumption recovery and prefer stocks like Kweichow Moutai.
>
>       The details of these two agent types are as below:
>
>       ```text
>       Agent O (As shown in Figure 5)
>       {
>         'Outline': 'A value-oriented investment approach focusing on fundamentally strong
>                     companies with a long-term perspective, preferring assets with low
>                     valuation and delivering stable and long-term return for holders',
>         'Details': {
>           'Risk Appetite': 'moderate',
>           'Holding Period': 'more than half a year',
>           'Strategy Consistency': '0.85',
>           'Rationality': '0.9',
>           'StockPoolSelector': 'IndustryEqualStockSelector',
>           'Others': 'A declining CPI growth rate and the government bond yields suggest
>                      that dividend assets may outperform.',
>           'Visible stocks': ['601088.SH', '600030.SH', '002594.SZ', ...]
>         }
>       }
>
>       Agent D (As shown in Figure 5)
>       {
>         'Outline': 'A value-oriented investment approach focusing on companies with
>                     stable and strong cash flows, preferring assets with lower
>                     valuation and higher profit quality',
>         'Details': {
>           'Risk Appetite': 'moderate',
>           'Holding Period': 'more than half a year',
>           'Strategy Consistency': '0.85',
>           'Rationality': '0.9',
>           'StockPoolSelector': 'IndustryEqualStockSelector',
>           'Others': 'Macroeconomic stimulus policies may lead to a recovery in consumption.',
>           'Visible stocks': ['600519.SH', '603259.SH', '000858.SZ', ...]
>         }
>       }
>       ```
>
>
>
> 3. During the market shift, dividend stocks outperformed. Our backward optimization mechanism detected this, increasing the weight of Agent O's strategy while decreasing Agent D's. This micro-level example demonstrates precisely how the framework adjudicates between competing, interpretable agent hypotheses to adapt its overall strategy.
>
> By combining the existing macro-level visualization with this new, detailed case study, we hope we can now offer a much clearer and more complete explanation of how agent interactions, mediated by our optimization process, directly drive portfolio performance.
>
> If you have any further questions or uncertainties, please feel free to raise them at any time. We are always happy to help!

---

> ### Author Response · Authors · 2025-11-17
>
> > **Question 1: Are there any limitations to the scaling effect, such as diminishing returns at very high numbers of agents? A more detailed analysis of potential trade-offs would be valuable.**
>
> This is an excellent and highly relevant question. Our paper's primary contribution was to discover and demonstrate this scaling effect up to 512 agents, but exploring its boundaries and trade-offs is critical.
>
> 1. **Diminishing Returns:** To investigate the limits of this effect, we conduct **new experiments on the SSE 50 index with 1024 and 1536 agents**. The results, now included in Appendix 3.4 and summarized below, confirm your hypothesis: performance continues to improve, but the rate of improvement begins to diminish. This is a classic characteristic of scaling effects.
>
>
> | NOA  | RIC  | RICIR | IC   | ICIR  |
> | :--- | :--- | :---- | :--- | :---- |
> | 512  | 8.16 | 41.74 | 5.90 | 33.43 |
> | 1024 | 9.25 | 43.02 | 6.27 | 34.19 |
> | 1536 | 9.22 | 43.11 | 6.29 | 34.05 |
>
> 1. **Analysis of Trade-offs:** The key trade-off is between marginal performance gain and computational cost.
>    1. **Performance:** It is worth noting that even with 512 agents, MASS achieves an annualized excess return of 12.14% compared to the SSE 50 (as shown in Appendix Table 5). This level of performance is particularly noteworthy as it approaches the upper threshold for index enhancement strategies on large-cap indices such as the SSE 50. It is achieved without relying on high-frequency features or employing market timing to manage portfolio exposure, placing our results in line with the Net Asset Value growth of top-tier private funds documented on [Simuwang](https://www.simuwang.com/), suggesting that MASS reaches a  competitive performance level well before the scaling effect completely plateaus. Therefore, we cautiously suggest that setting the total number of agents between 512 and 1024 offers a favorable trade-off.
>    2. **Cost:** As detailed in Table 6 in the Appendix, MASS is designed for online learning, so the daily computational cost is linear with the number of agents and the time complexity on each trading day is $ O(n_{type} \times n_{inv}) $. For a concrete example, the daily API cost for running MASS with 512 agents on the CSI 300 index is only about \$2.27. Even when scaling to a population of 1024 agents, the associated computational costs are negligible relative to the excess returns generated.
>
> In conclusion, the scaling effect remains evident as the Number of Agents (NOA) increases to 1024, though the improvement slows at an NOA of 1536. Notably, our MASS model already delivers strong performance at an NOA of just 512, with returns comparable to top-performing SSE 50 enhanced index products. Given the model's low execution costs, we find this performance to be highly compelling.

---

> ### Author Response · Authors · 2025-11-17
>
> We are grateful for your constructive feedback. We have worked to address your concerns in the revised paper and hope the changes are to your satisfaction. If you have any further questions, please feel free to contact us!

---

> ### Author Response · Authors · 2025-11-26
> **Request for Your Valuable Feedback**
>
> Dear Reviewer EMCk:
>
> We are very grateful for your thoughtful feedback and insights. With the Discussion Period set to officially end on December 2nd (AoE), we simply wanted to check in one last time to be certain that all your questions are fully addressed.
>
> We value your recognition and are always striving to improve our work.
>
> Best Regards,
>
> The Authors

---

> ### Author Response · Authors · 2025-11-28
>
> Dear Reviewer EMck,
>
> Thank you again for your invaluable time and expertise.
>
> In direct response to your points, our summary of changes is as follows: (1)We expanded our discussion to contextualize MASS against other multi-agent frameworks. (2)We validated its generalizability across different asset classes with experiments on the US markets. (3)To clarify agent interactions, we added a case study. （4）We explored the scaling effect's limits by testing with up to 1536 agents, confirming your hypothesis of diminishing returns and providing a thorough trade-off analysis.
>
> These additions and revisions have been incorporated into the updated manuscript.
>
> We hope our response and the revised paper could address your concerns. We would be grateful for your response and any further thoughts.
>
> Respectfully,
>
> The Authors

---

### Official Review · Reviewer_QTnN · 2025-11-01

**Soundness:** 3
**Presentation:** 3
**Contribution:** 2
**Rating:** 6
**Confidence:** 4

**Summary:**

The paper introduces MASS (Multi-Agent Simulation Scaling), an LLM-based framework for direct, end-to-end portfolio construction. Unlike previous models that rely on individual stock prediction or fixed workflows, MASS simulates a heterogeneous population of investor agents whose decisions are optimized through a novel backward optimization process. The system aggregates agent opinions based on the market disagreement hypothesis, balancing consensus and diversity, and demonstrates that increasing the number of agents (up to 512) leads to a scaling effect—higher returns and improved stability. Experiments on a self-collected 2023 Chinese A-share dataset show MASS outperforming seven state-of-the-art baselines across indices (SSE 50, CSI 300, ChiNext 100) and remaining robust under unseen 2025 data, suggesting strong adaptability and resistance to data leakage.

**Strengths:**

The paper makes a strong conceptual and empirical contribution to multi-agent financial modeling by reframing portfolio construction as a dynamic learning process rather than a prediction task. The backward optimization mechanism is novel and elegantly designed to simulate market adaptation, with clear empirical validation. The experiments are extensive—covering multiple indices, backtesting, ablation, and sensitivity studies. The scaling effect analysis is particularly impressive, revealing a near-linear improvement as agent count increases. The work’s open-sourced dataset and transparency on cost/time complexity further enhance its reproducibility and practical relevance.

**Weaknesses:**

Despite its innovation, the paper’s realism and generalizability could be questioned. The experiments rely exclusively on Chinese A-share data from 2023–2025, which may limit transferability to other markets. The backward optimization’s reliance on simulated annealing and hand-tuned hyperparameters introduces concerns about scalability and computational feasibility in live systems. Moreover, the LLM-driven agent logic depends heavily on textual prompts and lacks rigorous analysis of interpretability or reasoning validity. While the results are strong, there’s minimal theoretical justification for why the scaling effect should remain linear beyond the tested range, or how overfitting to historical macro features is prevented.

**Questions:**

How does MASS perform under longer backtesting horizons (e.g., multi-year or cross-market datasets)?

Can the backward optimization process be extended to reinforcement learning or differentiable optimization for efficiency?

How sensitive is MASS to the choice of LLM backbone (e.g., Qwen vs. GPT) given potential domain pretraining differences?

Is there a risk that the observed scaling benefit plateaus or reverses beyond 512 agents due to noise amplification?

---

> ### Author Response · Authors · 2025-11-17
>
> Thank you for your valuable time and insightful comments. We are delighted that you recognized the conceptual novelty and empirical contribution of our work. Your affirmation, coupled with your suggestions, has been instrumental in improving our manuscript. We will now address your concerns step by step.

---

> ### Author Response · Authors · 2025-11-17
>
> > **Weakness 1: Despite its innovation, the paper’s realism and generalizability could be questioned. The experiments rely exclusively on Chinese A-share data from 2023–2025, which may limit transferability to other markets.**
> >
> > **Question 1: How does MASS perform under longer backtesting horizons (e.g., multi-year or cross-market datasets)?**

---

> ### Author Response · Authors · 2025-11-17
>
> Thank you for your feedback concerning the realism of our experiments and the generalizability of our findings beyond the Chinese A-share market.
>
> **On Realism:** We understand your concern and have made every effort to ensure our work is transparent and reproducible. We have secured permission from the company to open-source all code, data, and intermediate experimental results. To this end, we have fully open-sourced the MASS framework, including:
>
> - The complete source code: `[https://anonymous.4open.science/r/MASS-AC96`] or `[https://github.com/anonymous3728/MASS_anonoymous`]
> - The comprehensive Chinese A-share SSE 50 dataset used in our study, which includes firm-level features, news, financial reports, and macroeconomic indicators: `[https://anonymous.4open.science/r/MASS-AC96/stock_disgareement/dataset/]` or `[https://github.com/anonymous3728/MASS_anonoymous/stock_disagreement/dataset`]
> - Training snapshots containing the agent distribution on the SSE 50 Index, allowing for reproductivity of our findings: `[https://anonymous.4open.science/r/MASS-AC96/ih_dist]` or `[https://github.com/anonymous3728/MASS_anonoymous/stock_disagreement/ih_dist`]
>
> We sincerely hope that our commitment to full transparency and reproducibility could alleviate any doubts regarding the realism of our results.
>
> **On Generalizability:** We agree that demonstrating generalizability is crucial. We have addressed this in two ways:
>
> - **Diverse Stock Pools (Intra-Market):** As detailed in the paper, our experiments already cover three distinct Chinese indices: the **SSE 50** (large-cap value stocks), **CSI 300** (broad market), and **ChiNext 100** (growth-oriented tech stocks). The strong performance of MASS across these stylistically different pools provides initial evidence of its ability to generalize beyond a single type of asset.
>
> - **New Experiments on US Markets (Cross-Market):** To directly address your concern about transferability to other markets, we conduct new experiments on the **Nasdaq 100** and **S&P 500** indices for the same 2023 and 2025Q1 periods in the paper. The data was sourced from public repositories like Microsoft Qlib and Yahoo Finance. The results, shown below and detailed in Table 8 in the Appendix, show that MASS continues to perform effectively across different asset classes compared to various SOTA baselines, confirming its robustness in a different market environment.
>
>   Main experiments on Nasdaq-100 and S&P 500 (2023).  All values are in percent. We also add two new SOTA baselines:  FactorVAE [1] and HireVAE [2] , representing the factor model.
>
> | Method                                     | Nasdaq-100 RIC | Nasdaq-100 RICIR | Nasdaq-100 IC | Nasdaq-100 ICIR | S&P 500 RIC | S&P 500 RICIR | S&P 500 IC | S&P 500 ICIR |
> | ------------------------------------------ | -------------: | ---------------: | ------------: | --------------: | ----------: | ------------: | ---------: | -----------: |
> | **Proxy Indicator (Diether et al., 2002)** |           1.94 |            15.37 |          1.82 |           13.91 |        1.85 |         16.02 |       1.93 |        14.31 |
> | **LightGBM (2017)**                        |           2.71 |            19.90 |          2.56 |           19.34 |        2.06 |         19.84 |       2.19 |        17.83 |
> | **FactorVAE (2022)**                       |           3.49 |            26.05 |          3.62 |       **28.95** |        3.96 |         28.34 |       3.77 |        29.64 |
> | **HireVAE(2023)**                          |           3.52 |            25.30 |          3.79 |           27.98 |        4.12 |         27.86 |       3.83 |        28.39 |
> | **DTML (2021)**                            |           3.15 |            22.90 |          2.83 |           21.56 |        3.52 |         24.65 |       2.96 |        20.10 |
> | **MASTER (2024)**                          |           3.38 |            23.62 |          2.98 |           21.49 |        3.27 |         25.93 |       3.09 |        22.53 |
> | **SEP (2024)**                             |           3.40 |            22.99 |          3.26 |           23.85 |        1.38 |         11.82 |       0.82 |         7.81 |
> | **FinCON (2024)**                          |           3.46 |            23.81 |          3.24 |           24.77 |        1.24 |         10.27 |       0.68 |         8.64 |
> | **TradingAgents (2025)**                   |           3.63 |            27.36 |          3.85 |           28.29 |        4.07 |         31.28 |       3.89 |        27.94 |
> | **MASS**                                   |       **4.27** |        **31.05** |      **3.94** |           28.90 |    **4.31** |     **31.45** |   **3.95** |    **28.68** |

---

> ### Author Response · Authors · 2025-11-17
>
> Data Leakage experiments on Nasdaq and S&P 500 (Q1 2025)
>
> | Method                                     | Nasdaq-100 RIC | Nasdaq-100 RICIR | Nasdaq-100 IC | Nasdaq-100 ICIR | S&P 500 RIC | S&P 500 RICIR | S&P 500 IC | S&P 500 ICIR |
> | ------------------------------------------ | -------------: | ---------------: | ------------: | --------------: | ----------: | ------------: | ---------: | -----------: |
> | **Proxy Indicator (Diether et al., 2002)** |           1.98 |            17.26 |          1.47 |           14.83 |        2.06 |         16.39 |       2.34 |        15.81 |
> | **LightGBM (2017)**                        |           2.40 |            18.75 |          2.38 |           19.36 |        2.64 |         19.42 |       2.47 |        17.38 |
> | **FactorVAE (2022)**                       |           3.42 |            27.86 |          3.29 |           27.05 |        3.55 |         24.60 |       3.49 |        27.85 |
> | **HireVAE(2023)**                          |           3.58 |            24.97 |          3.63 |           26.37 |        3.67 |         24.54 |       3.72 |        27.63 |
> | **DTML (2021)**                            |           3.21 |            23.59 |          2.93 |           21.40 |        3.37 |         22.35 |       3.26 |        21.84 |
> | **MASTER (2024)**                          |           3.52 |            25.98 |          3.20 |           25.84 |        3.61 |         26.54 |       3.48 |        25.70 |
> | **SEP (2024)**                             |           3.43 |            26.35 |          3.19 |           25.76 |        0.62 |          6.35 |       0.74 |         5.89 |
> | **FinCON (2024)**                          |           3.48 |            25.82 |          3.63 |           25.97 |        1.13 |          8.56 |       0.97 |         6.75 |
> | **TradingAgents (2025)**                   |           3.50 |            26.76 |          3.71 |           26.99 |        3.78 |         28.04 |       3.92 |        29.31 |
> | **MASS**                                   |       **3.96** |        **29.84** |      **4.01** |       **27.53** |    **4.05** |     **29.73** |   **3.99** |    **29.67** |
>
> We hope that these new cross-market experiments, combined with our results on diverse Chinese indices, can provide compelling evidence for the generalizability of the MASS.
>
> References:
>
> [1] Duan, Yitong, et al. "Factorvae: A probabilistic dynamic factor model based on variational autoencoder for predicting cross-sectional stock returns." AAAI 2022.
>
> [2] Wei, Zikai, et al. "Hirevae: An online and adaptive factor model based on hierarchical and regime-switch vae." IJCAI 2023.

---

> ### Author Response · Authors · 2025-11-17
>
> > **Weakness 2: The backward optimization’s reliance on simulated annealing and hand-tuned hyperparameters introduces concerns about scalability and computational feasibility in live systems.**
>
> Thank you for raising this point about the practical implementation of our backward optimization process.
>
> **On Scalability and Hyperparameter Sensitivity:** In the original submission (Figure 3), we provided a sensitivity analysis for the core hyperparameter \( $\alpha$ & $\omega_{opt}$). The experiment results show that hyperparameters lead to only slight variations in system performance.
>
> To further address your concern, we conducted an additional sensitivity analysis on the hyperparameters of the simulated annealing optimizer, specifically the cooling rate and the number of iterations. This analysis uses the same experimental setup described in Section 4.1.7 of our original submission.
>
> These new results, presented in Table 4 in the Appendix and below, show that MASS's performance is robust across a reasonable range of these parameters. The results are twofold. The optimizer demonstrates considerable robustness to hyperparameter tuning. For the cooling rate, values within the 0.85–0.98 range yielded consistent and strong results; optimization failure occurred only upon the complete omission of the cooling schedule. Similarly, the model reliably converged with over 100 iterations, indicating a wide tolerance for this parameter. Non-convergence was observed only with an insufficient number of iterations. In light of these findings, we cautiously suggest that our optimizer is not overly sensitive to its hyperparameter settings.
>
>
> | Cooling rate |      RIC |     RICIR |       IC |      ICIR |
> | ------------ | -------: | --------: | -------: | --------: |
> | 1.00         |    -0.16 |     -3.58 |    -0.27 |     -4.99 |
> | 0.98         |     5.79 |     39.68 |     4.21 |     30.81 |
> | 0.95         |     6.50 |     43.49 |     4.65 |     33.32 |
> | 0.90         | **6.53** | **44.82** | **4.77** | **34.06** |
> | 0.85         |     5.81 |     40.12 |     4.16 |     31.13 |
> | 0.80         |     4.12 |     31.90 |     3.89 |     24.58 |
>
> | Max iteration |      RIC |     RICIR |       IC |      ICIR |
> | ------------- | -------: | --------: | -------: | --------: |
> | 0             |     0.36 |      5.36 |     0.41 |      6.69 |
> | 25            |     3.04 |     23.55 |     2.94 |     21.89 |
> | 50            |     4.69 |     31.80 |     3.73 |     26.66 |
> | 100           |     6.50 | **43.49** |     4.65 | **33.32** |
> | 200           | **6.53** |     42.76 | **4.66** |     32.91 |
>
> **On Computational Feasibility:** We would like to clarify that MASS does not need high computation costs. On any given day \( j \), we only need to perform one single forward pass and one backward optimization step. In each backward pass, the parameters of the agent distribution are iteratively optimized until convergence, using historical returns of all assets in the stock pool from the optimization window (as defined in Section 3.2). In each backward pass, the parameters of the agent distribution are iteratively optimized until convergence, using historical returns of all assets in the stock pool from the optimization window, as defined in Section 3.2 in the original submission. By caching agent decisions from previous days, the daily computational overhead is therefore only $O(n_{type} \times n_{inv})$. To offer a concrete estimate, as detailed in Table 6 in the Appendix in the original submission, **the daily API costs for updates on the SSE 50, CSI 300, and ChiNext 100 indices are merely approximately \$0.68, \$2.27, and \$1.19, respectively (based on the Qwen2.5 72B API pricing in the Alibaba Cloud platform)**. We believe this manageable daily cost makes MASS highly feasible for both independent researchers and financial institutions.
>
> We hope that our existing parameter-sensitivity experiments, the additional ones conducted per your request, and our analysis of computational feasibility address your concerns. If you have any further questions or doubts, please feel free to contact us at any time. We will be glad to solve it for you.

---

> ### Author Response · Authors · 2025-11-17
>
> >**Weakness 3: Moreover, the LLM-driven agent logic depends heavily on textual prompts.**
>
>
>
> Thank you for raising this question regarding the role of textual prompts and the potential for prompt-dependency in our framework.
>
> We would like to clarify a key aspect of our design: **MASS is intentionally engineered to be robust and** ***not*** **dependent on specific, finely-tuned prompts.** The system's strength comes from its architecture, not from fragile prompt engineering. This is achieved in several ways:
>
> First, as detailed in Section 3.1.1, the system's heterogeneity is generated programmatically. The agent types are created based on combinations of investment archetypes, and the subset of stocks each agent instance can "see" is assigned randomly. We do not hand-craft prompts for each of the 512 agent instances. Instead, we use a generalized template that only provides the necessary context: the agent's persona, its available information, and its objective. A typical prompt template to generate agent types in Section 3.1.1 looks like this:
>
>
>
> ```text
> # original prompt
>
> Give the following input data:
> Input time-series data column name and their descriptions in JSON format(textual data example).
> latest macroeconomic and market insights.
> Please try to analyze and summarize an abstract investing style description.
> The output format is a json. The specific format of the output JSON is:
> {
>     "Outline": "The outline and general description for investment style within 50 words. The outline is a summarization about your investing strategy and your insights into the subsequent trend of the stock market, without any details below.",
>     "Details": {
>         "Risk Appetite": "conservative | moderate | moderately conservative | moderately aggressive | aggressive",
>         "Holding Period": "one day | about one week | about one month | about half a year | more than one year",
>
>         "Strategy Consistency": [0, 1] (Refers to the investor's ability to adhere to and execute their investment strategy with persistence and coherence, regardless of short-term market fluctuations or emotional influences. Higher number means high consistency",
>
>         "Rationality": [0, 1] (Refers to whether the investor's decision-making process is based on logic, data, and long-term objectives rather than emotions, biases, or short-term market noise. Higher number means high rationality",
>
>         "StockPoolSelector": "Specify what kind of preference you'd like to construct your watchlist stocks. The possible preferences are:
>
>   1. RandomStockSelector: Randomly construct your watchlist.
>
>   2. IndustryEqualStockSelector: Construct a stock pool with balanced distribution across industries.
>
>   3. MVEqualStockSelector: Construct a stock pool with balanced distribution across market capitalizations.
>
>   4.IndustryBasisStockSelector: Prefer stocks from specific industries and output the preferred industries. The result is presented in a list format.",
>
>         "Others": "Extra information about your investing strategy, maybe correlated with latest market and macroeconmic information and others. No more than 30 words."
>     }
> }
>
> {examples}
>
> Input data:
> {input_data}
> ```

---

> ### Author Response · Authors · 2025-11-17
>
> Second, to empirically validate this low sensitivity to prompt phrasing, we conducted an additional experiment. We significantly rephrased the instructional part of the prompt for all agent initialization part in the Section 3.1.1  —altering the wording, sentence structure, and tone—and re-ran the simulation on the SSE 50 index.
>
> ```
> # changed prompt
>
> Task: Generate Investment Profile JSON
> OBJECTIVE: Synthesize a structured investment profile by analyzing provided data schema and market intelligence.
> INPUT PAYLOAD:
> data_schema: A JSON object defining available time-series data, including column names and descriptions.
> market_intelligence: A textual summary of the latest macroeconomic and market insights.
> CONSTRAINTS:
> The output MUST be a single, valid JSON object that strictly adheres to the following structure and constraints. No additional text or explanation is permitted outside the JSON structure.
>
> {
>     "Outline": "string // Max 50 words. A high-level summary of the investment strategy and market outlook. Must not contain details found in the 'Details' object.",
>     "Details": {
>         "Risk Appetite": "string // ENUM: Must be one of 'conservative', 'moderate', 'moderately conservative', 'moderately aggressive', 'aggressive'.",
>         "Holding Period": "string // ENUM: Must be one of 'one day', 'about one week', 'about one month', 'about half a year', 'more than one year'.",
>         "Strategy Consistency": "float // RANGE: [0.0, 1.0]. Represents adherence to strategy despite market noise. 1.0 is maximum consistency.",
>         "Rationality": "float // RANGE: [0.0, 1.0]. Represents logic-driven vs. emotional decision-making. 1.0 is maximum rationality.",
>         "StockPoolSelector": "string // ENUM: Must be one of 'RandomStockSelector', 'IndustryEqualStockSelector', 'MVEqualStockSelector', 'IndustryBasisStockSelector'. If 'IndustryBasisStockSelector' is chosen, specify preferred industries in a list.",
>         "Others": "string // Max 30 words. Supplementary notes on strategy, potentially linking to the micro-economic information and your insights into strategies."
>     }
> }
>
> {examples}
>
> Input data:
> {input_data}
> ```
>
> The performance still remained strong in the SSE 50 Index. As shown below:
>
> |            | RIC  | RICIR | IC   | ICIR  |
> | :--------- | :--- | :---- | :--- | :---- |
> | **BEFORE** | 8.16 | 41.74 | 5.90 | 33.43 |
> | **AFTER**  | 8.09 | 41.75 | 5.94 | 34.13 |
>
> This experiment demonstrates that the framework's success is not an artifact of specific prompt wording. Instead, it underscores our central thesis: the power of MASS lies not in the linguistic formulation for any single agent, but in the emergent intelligence derived from aggregating diverse agent opinions and the adaptive learning driven by the backward optimization process. This automated optimization, which dynamically learns to weigh different agent strategies based on market feedback, is the core innovation that drives the system's performance and adaptability.
>
> We hope that our explanations and additional experiments can address your concerns. If you have any further questions, please feel free to contact us.

---

> ### Author Response · Authors · 2025-11-17
>
> >**Weakness 4: Moreover, the LLM-driven agent logic lacks rigorous analysis of interpretability or reasoning validity.**
>
>
>
> We appreciate your feedback on the need for a more analysis of the agents' reasoning processes.
>
> First, we would like to draw your attention to Figure 5 in the paper, which provides a macro-level view of the system's interpretability. This visualization shows how the aggregate distribution of agent types dynamically shifts in response to major market regime changes (marked as A and B). This demonstrates that the backward optimization process is successfully learning and adapting the system's collective strategy to align with evolving market dynamics.
>
> To provide the granular, micro-level view you suggested, we have added a new case study in Appendix A.3.5. For your convenience and to offer full transparency into the agents' reasoning, we present the complete prompts for two key agents from this case study below.
>
>
>
> ```text
> Agent O (As shown in Figure 5)
> {
>   'Outline': 'A value-oriented investment approach focusing on fundamentally strong
>               companies with a long-term perspective, preferring assets with low
>               valuation and delivering stable and long-term return for holders',
>   'Details': {
>     'Risk Appetite': 'moderate',
>     'Holding Period': 'more than half a year',
>     'Strategy Consistency': '0.85',
>     'Rationality': '0.9',
>     'StockPoolSelector': 'IndustryEqualStockSelector',
>     'Others': 'A declining CPI growth rate and the government bond yields suggest
>                that dividend assets may outperform.',
>     'Visible stocks': ['601088.SH', '600030.SH', '002594.SZ', ...]
>   }
> }
>
> Agent D (As shown in Figure 5)
> {
>   'Outline': 'A value-oriented investment approach focusing on companies with
>               stable and strong cash flows, preferring assets with lower
>               valuation and higher profit quality',
>   'Details': {
>     'Risk Appetite': 'moderate',
>     'Holding Period': 'more than half a year',
>     'Strategy Consistency': '0.85',
>     'Rationality': '0.9',
>     'StockPoolSelector': 'IndustryEqualStockSelector',
>     'Others': 'Macroeconomic stimulus policies may lead to a recovery in consumption.',
>     'Visible stocks': ['600519.SH', '603259.SH', '000858.SZ', ...]
>   }
> }
> ```
>
> As these prompts illustrate, each agent operates based on a distinct and interpretable mandate. During the market regime shift B, their reasoning diverges:
>
> - **Agent O**, guided by the logic in its `'Others'` field, identifies a macro environment favorable to dividend assets ("*dividend assets may outperform*") and thus favors China Shenhua Energy (`601088.SH`).
> - **Agent D**, following its own directive, anticipates a "*recovery in consumption*" due to stimulus policies, leading it to prefer Kweichow Moutai (`600519.SH`).
>
> This creates two competing, well-reasoned hypotheses. Our backward optimization mechanism then adjudicates between them by analyzing their performance in the lookback window. By observing the outperformance of the dividend-focused strategy during the shift, the mechanism **increases the allocation to Agent O while reducing that of Agent D**.
>
> This case study offers concrete evidence for how our framework operates: it leverages diverse, human-understandable agent reasoning and then uses a data-driven optimization process to dynamically select the most effective strategies. We hope this detailed walk-through addresses your concerns about the validity and interpretability of the agent reasoning process.

---

> ### Author Response · Authors · 2025-11-17
>
> >**Weakness 5: While the results are strong, there’s minimal theoretical justification for why the scaling effect should remain linear beyond the tested range, or how overfitting to historical macro features is prevented.**
> >
> >**Question 4: Is there a risk that the observed scaling benefit plateaus or reverses beyond 512 agents due to noise amplification?**
>
>
>
> Thank you for this thoughtful question about the limits of the scaling effect and the potential for overfitting.
>
> **On the Scaling Effect:** We acknowledge that we cannot theoretically prove that the observed linear scaling effect will continue indefinitely. Our primary contribution is the empirical discovery and validation of this powerful effect within a previously unexplored range (up to 512 agents). To push this boundary further, we have run **new experiments on the SSE 50 index with 1024 and 1536 agents**, with the results added to Appendix 3.4 and below. The results show that performance continues to improve, although the rate of improvement begins to slightly diminish, which is a common characteristic of scaling effects. We believe this is analogous to other foundational scaling effect papers, where the contribution lies in demonstrating the effect's existence and utility within a practical range, rather than proving its infinite extension. We hope you would recognize the potential our work reveals.
>
> | NOA  | RIC  | RICIR | IC   | ICIR  |
> | :--- | :--- | :---- | :--- | :---- |
> | 512  | 8.16 | 41.74 | 5.90 | 33.43 |
> | 1024 | 9.25 | 43.02 | 6.27 | 34.19 |
> | 1536 | 9.22 | 43.11 | 6.29 | 34.05 |
>
> We acknowledge that our MASS shows signs of a performance bottleneck on the SSE 50 at 1,536 agents. However, we would like to humbly offer some context for this finding. According to the private fund statistics platform [Simuwang](https://www.simuwang.com/), the practical limit for index enhancement strategies on major indices (without employing  high frequency trading and holding position timing strategies) with high market value(such as the SSE 50)  is an annualized excess return of around 12-15%. We were therefore encouraged to see that, as shown in Table 5 and Table 7 of the Appendix, MASS with 512 agents had already delivered a 12.14% (Table 7 2.16% ARR on MASS and -9.98% on the SSE 50) excess return. This suggests our method achieves highly competitive performance well before reaching its computational scaling limits.
>
> **On Overfitting to Macro Features:** We have evidence to mitigate this concern:
>
> 1. **LLM Knowledge Cut-off:** The `Qwen-2.5-72B-Instruct` model has a knowledge cut-off prior to 2025. As shown in Table 1, MASS maintains its strong performance on new, unseen data from Q1 2025. This demonstrates that the system is performing genuine reasoning on novel information, not simply recalling memorized patterns from its training data.
> 2. **Cross-Market Generalization:** As presented in our response to Weakness 1, our new experiments show that MASS generalizes effectively to the US market (Nasdaq 100 and S&P 500), which operates under a completely different macroeconomic context than the Chinese A-share market. This would be highly unlikely if the model were merely overfitting to specific macro features of a single economy.
>
> Regarding Question 4, we hypothesize that our theory-grounded aggregation method (Eq. 4), which explicitly balances consensus and penalizes disagreement, acts as a natural noise filter. A performance reversal would likely only occur if the marginal information from new agents becomes pure noise, which does not appear to be the case within the ranges we tested. The framework effectively extracts signal from a larger pool of agents without being overwhelmed by noise.
>
>
>
> We hope the additional experiments and our explanation of MASS's market position could address your concerns. Should you have any further questions, please do not hesitate to ask. We are always happy to answer them.

---

> ### Author Response · Authors · 2025-11-17
>
> > **Question 2: Can the backward optimization process be extended to reinforcement learning or differentiable optimization for efficiency?**
>
> This is a fantastic suggestion for future research, touching on two very promising paths to enhance the optimization core of MASS.
>
> **Reinforcement Learning:** Our current framework can be naturally reframed as a policy optimization problem. One could design an RL agent where the "action" is to select the investor distribution $ d_t $ for the next trading day. The "state" would be a representation of recent market conditions and the agents' historical decisions, and the "reward" would be the subsequent portfolio return or Rank IC. This would replace the fixed-look-back-window optimization of simulated annealing with a continuous learning policy that could, in theory, learn more complex temporal dependencies and adapt even more dynamically to market regime shifts.
>
> **Differentiable Optimization:** In addition to RL, pursuing differentiable optimization is another highly promising direction for improving efficiency. The core idea would be to construct a fully differentiable computation graph from the agent distribution  $ d_t $ to the final performance metric. This would allow us to replace the derivative-free, search-based simulated annealing with far more efficient gradient-based optimizers like Adam. However, this presents two primary challenges:
>
> 1. Non-Differentiable Agent Outputs: The LLM-based agents currently produce discrete outputs (a list of selected stocks), which breaks the gradient flow.
> 2. Non-Differentiable Objective: Our objective function, the Rank Information Coefficient (RIC), is also non-differentiable due to its reliance on sorting.
>
> To overcome these hurdles, one could re-engineer the agents' task to have them output continuous values, such as probabilities or logits for each stock in their pool. This would enable the use of techniques like the Gumbel-Softmax trick to create a continuous and differentiable approximation of the discrete selection process. Simultaneously, one could replace RIC with a differentiable proxy metric, such as the standard IC or a simpler portfolio return objective. While this approach promises significant gains in optimization speed, it involves a trade-off in architectural complexity and potential approximation errors. Investigating this path is a compelling direction for future work.

---

> ### Author Response · Authors · 2025-11-17
>
> > **Question 3: How sensitive is MASS to the choice of LLM backbone (e.g., Qwen vs. GPT) given potential domain pretraining differences?**
>
>
>
> This is a good point. To test the framework's sensitivity to the underlying LLM, we have conducted a new set of experiments on the China stock market using GPT-OSS-120B  as the backbone with the knowledge cutoff in June 2024. The comparative results are presented below and have been added to Table 1 and Table 7. The detailed results are below:
>
> Main Experiments on 2023
>
>
>
> | Method                  | Dataset          | RIC      | RICIR     | IC       | ICIR      | ARR      | SR       | MDD       |
> | :---------------------- | :--------------- | :------- | :-------- | :------- | :-------- | :------- | :------- | :-------- |
> | **MASS (Qwen)**         | **SSE50**        | 8.16     | 41.74     | 5.90     | **33.43** | **2.16** | 1.98     | 11.98     |
> | **MASS (GPT-OSS-120B)** | **SSE50**        | **8.24** | **41.96** | **5.91** | 33.28     | 2.14     | **1.99** | **11.36** |
> | **MASS (Qwen)**         | **CSI 300**      | 6.50     | **43.49** | **4.65** | **33.32** | **4.95** | **2.23** | **14.04** |
> | **MASS (GPT-OSS-120B)** | **CSI 300**      | **6.62** | 41.96     | 4.63     | 30.19     | 4.87     | 2.06     | 14.87     |
> | **MASS (Qwen)**         | **Chi Next 100** | 7.62     | **62.87** | 6.28     | **55.88** | 1.17     | **0.99** | **19.06** |
> | **MASS (GPT-OSS-120B)** | **Chi Next 100** | **7.66** | 61.56     | **6.43** | 54.29     | **1.26** | 0.97     | 22.67     |
>
> Experiments on Data Leakage Concern (The first quarter of 2025)
>
>
> | Method                  | Dataset      | RIC      | RICIR     | IC       | ICIR      | ARR       | SR       | MDD      |
> | :---------------------- | :----------- | :------- | :-------- | :------- | :-------- | :-------- | :------- | :------- |
> | **MASS (Qwen)**         | **SSE50**    | 4.50     | 24.41     | 6.12     | **38.33** | 9.74      | **2.42** | **2.91** |
> | **MASS (GPT-OSS-120B)** | **SSE50**    | **4.56** | **24.56** | **6.31** | 37.98     | **9.81**  | 2.38     | 3.04     |
> | **MASS (Qwen)**         | **CSI 300**  | **3.91** | **37.44** | **3.36** | **34.56** | **9.36**  | **2.66** | **2.99** |
> | **MASS (GPT-OSS-120B)** | **CSI 300**  | 3.75     | 35.86     | 3.31     | 33.80     | 8.42      | 2.49     | 3.04     |
> | **MASS (Qwen)**         | **CSI A500** | 5.19     | **56.17** | 4.66     | **48.82** | 11.34     | **2.93** | **4.08** |
> | **MASS (GPT-OSS-120B)** | **CSI A500** | **5.27** | 54.72     | **4.68** | 46.05     | **11.51** | 2.88     | 4.17     |
>
> The results show that while there is a slight performance difference, MASS with GPT-OSS-120B still  gains enhanced performance. This strongly suggests that the success of MASS is rooted in its architectural design—the multi-agent simulation and backward optimization, rather than being dependent on a single, specific LLM.

---

> ### Author Response · Authors · 2025-11-17
>
> Thank you for your valuable feedback and insights. I sincerely hope that our responses above have helped clarify any concerns you may have. We would be very grateful if you could carefully consider your final rating. It is extremely important to us. Please feel free to reach out with any additional questions or comments. We would be more than happy to assist further.

---

> ### Author Response · Authors · 2025-11-26
> **Kindly Requesting Reviewer Feedback**
>
> Dear Reviewer QTnN:
>
> Thank you so much for sharing your valuable feedback and insights with us. Note that the Discussion Period will be concluded on December 2nd, AoE, we just wanted to gently follow up and ensure that our responses were sufficient and that all your questions have been thoroughly resolved.
>
> We truly appreciate your time and hope to earn your further recognition for our work.
>
> Best Regards,
>
> The Authors

---

> > ### Author Response · Authors · 2025-11-28
> >
> > Dear Reviewer QTnN,
> >
> > Thank you once again for your thoughtful review.
> >
> > In direct response to your points, our summary of changes is as follows: (1)We addressed the concerns about generalizability by conducting experiments on the US markets.  (2)We provided sensitivity analyses on optimizer hyperparameters and prompt phrasing. (3) We added a case study for interpretability.(4)We tested the scaling effect with up to 1536 agents and validated our framework using a different LLM backbone.
> >
> > These new results have been incorporated into the revised manuscript.
> >
> > As the ICLR discussion period is drawing to a close, we wonder if our revisions and clarifications have addressed your concerns. We would be grateful if you could spare a moment to provide any further thoughts.
> >
> > Sincerely,
> >
> > Authors

---

### Author Response · Authors · 2025-11-17

Dear Reviewers, ACs:

We thank all reviewers for their valuable time, insightful comments, and constructive feedback. Your efforts have been instrumental in helping us strengthen our manuscript.

We are encouraged that the reviewers recognized the novelty and contributions of our work. We have summarized the highlighted strengths below:

- **Conceptual Novelty:** Reviewers praised our work for its "strong conceptual and empirical contribution" (QTnN) and for reframing portfolio construction as a dynamic learning process. The core idea was found to be "conceptually appealing" (xnHx), and the framework's design was described as "vivid" (ym2m). Reviewer EMCk appreciated the "originality" of viewing our framework as a "new approach to direct portfolio construction."
- **Methodological Strength:** The backward optimization mechanism was described as "novel and elegantly designed" (QTnN). Reviewers also noted that this is the "first work that explores the influence of scales when leveraging multi-agent settings for stock trading" (ym2m).
- **Empirical Rigor & Evaluation:** Our experiments were deemed "extensive" (QTnN) and "solid, with varied SoTA baseline methods" (ym2m). The "scaling effect analysis is particularly impressive" (QTnN), and the "Ablation study is conducted to sufficiently justify the choice of each component" (ym2m).
- **Reproducibility & Openness:** We are glad reviewers appreciated that the "release of code and dataset enhances openness and encourages further research" (xnHx) and that this transparency "further enhance[s] its reproducibility and practical relevance" (QTnN).

We have categorized the reviewers' concerns and suggestions below, along with a summary of how we addressed them.

**1. Concerns about Generalizability and Realism**

- **Concern:** The experiments were limited to the Chinese A-share market, raising questions about generalizability to other markets and asset classes (QTnN, EMCk).
- **Our Response:** To directly address this, we conducted new experiments on the US market, evaluating MASS and all baselines on the **Nasdaq 100 and S&P 500** indices for both the 2023 and Q1 2025 periods. The results, added to Table 8 in the Appendix, show that MASS maintains its enhanced performance in a completely different market environment, confirming its robustness and generalizability.

**2. Concerns about Methodological Soundness and Feasibility**

- **Concern:** The use of Simulated Annealing (SA) was questioned for its computational feasibility and consistency with a dynamic online learning framework (QTnN, xnHx).
- **Our Response:** We clarified that MASS operates on a daily frequency, where the sub-10-minute optimization time is highly practical. To further demonstrate robustness, we conducted a **new sensitivity analysis** on SA's hyperparameters (cooling rate and number of iterations), showing that performance is relatively stable across a reasonable range of settings.
- **Concern:** The framework's success might depend heavily on prompt engineering, and the agent reasoning process lacked interpretability (QTnN, ym2m).
- **Our Response:** We explained that agent diversity is programmatically generated, not manually prompt-engineered. To prove this, we conducted a **new experiment where we radically rephrased the core prompts** and showed that performance remained stable. For interpretability, we added **two  new, detailed case studys** in Appendix 3.5, providing full agent prompts and a micro-level walkthrough of how the system adapts to a market regime shift.
- **Concern:** The relationship between MASS and traditional factor models was unclear, and key factor model baselines were missing (xnHx).
- **Our Response:** We thank the reviewer for this suggestion. We have now **added two state-of-the-art factor models (FactorVAE and HireVAE)** as baselines across all of our experiments. This includes four new sets of comparisons: on the Chinese market (2023 & 2025), and the US market (2023 & 2025). The results show MASS outperforming these strong baselines, demonstrating that our agent-based simulation captures information beyond conventional factors.
- **Concern:** The paper lacked a theoretical justification for the scaling effect's limits (QTnN, EMCk).
- **Our Response:** We extended our empirical validation. We ran **new experiments scaling MASS to 1024 and 1536 agents**. The results, which show diminishing but still positive returns,  are crucial for understanding the trade-off between performance gains and the onset of performance saturation at larger scales.

---

> ### Author Response · Authors · 2025-11-17
>
> **3. Suggestions for Additional Experiments and Analysis**
>
> - **Concern:** The data leakage experiment for Q1 2025 was missing performance data for the baseline models (ym2m).
> - **Our Response:** We have now **run all baseline models on the Q1 2025 data** for both the Chinese and US markets. The complete results have been added to the paper, providing a full and fair comparison that validates MASS's consistent outperformance on unseen data.
> - **Concern:** The framework's sensitivity to the choice of the underlying LLM was not tested (QTnN).
> - **Our Response:** We conducted a **new set of experiments replacing the Qwen backbone with GPT-OSS-120B**. The results show that MASS maintains its strong performance, indicating that the framework's success is rooted in its architecture rather than dependence on a specific LLM.
> - **Concern:** The interpretation of the Market Disagreement Hypothesis (MDH) seemed inconsistent (xnHx).
> - **Our Response:** We clarified our methodology and added more explanations.  To empirically support our explanations, we ran a **new ablation study** comparing our `Consensus - Disagreement` formula against alternatives, empirically confirming our approach.
>
> We are grateful for the opportunity to improve our paper with this feedback. We hope the addition of multiple new cross-market experiments, new baselines, new hyperparameter sensitivity analysis and ablation study, and deeper analyses can strengthen the manuscript and address the reviewers' concerns.
>
> For our detailed/point-by-point responses to each reviewer, please see the respective threads.

---

### Author Response · Authors · 2025-11-25
**Revision Summary**

Dear Reviewers:

We are sincerely grateful for your insightful comments and suggestions. We have carefully addressed all points and revised the manuscript accordingly. For your convenience, we summarize the major revisions below.

## Reviewer QTnN

1. **The paper’s realism and generalizability :**
   * Section 4 Dataset and stock pools (Line318-320)
   * Section 4.1.1 Main Experiments (Line 360-365)
   * Section 4.1.2  Experiments on Data Leakage concern (Line 373)
   * Appendix A.5.2 Results on the US stock market (Line 1346-1349); Table 8 (Line 1296-1327)
   * A fully anonymized code link backup: Page 1 footnote (Line 53)
2. **The  hyperparameter of backward optimization's potential effect on sensitivity and computational feasiblity.**
   * Section 4.1.7  Parameter Sensitivity Experiments (Line515 - 517)
   * Appendix A.3.6 More Parameter Sensitivity Analysis (Line1149 - 1155)
   * Table 4 (Line 1188-1209)
3. **Analysis of interpretability or reasoning validity.**
   * Section 4.1.6: stability and agent distribution experiment (Line 481-485)
   * Update on Figure 5 (Line 486-488)
   * Appendix3.5: Case Study (Line 1038-1072)
4. **More investigation on the scaling effect.**
   * Section 4.1.4: Scaling Experiments (Line 429-431)
   * Appendix A.3.4: More discussions on scaling experiments (Line 1029-1036)
   * Table 5(Line 1210-1217)
5. **Sensitivity of MASS to the choice of LLM backbone.**
   * Section 4 Experiment specific settings (Line 335-336)
   * Section 4.1.1: Main Experiments (Line 365)
   * Update on Table 1 (Line 293,304)
   * Update on Table 7 (Line 1259, 1275)



## Reviewer EMck

1. **Comparison to other multi-agent simulation approach in finance; exploring different asset classes or regimes**.

   * Section 2.3: scaling effects in multi-agent systems (Line 144-156)
   * Section 4 Dataset and stock pools (Line318-320)
   * Section 4.1.1 Main Experiments (Line 360-365)
   * Section 4.1.2  Experiments on Data Leakage concern (Line 373)
   * Appendix A.5.2 Results on the US stock market (Line 1346-1349); Table 8 (Line 1296-1327)

2. **The interaction between agents and their impact on portfolio performance .**

   * Section 4.1.6: stability and agent distribution experiment (Line 481-485)

   * Update on Figure 5 (Line 486-488)
   * Appendix3.5: Case Study (Line 1038-1072)

3. **More investigation on the scaling effect.**

   * Section 4.1.4: Scaling Experiments (Line 429-431)
   * Appendix A.3.4: More discussions on scaling experiments (Line 1029-1036)
   * Table 5 (Line 1210-1217)



## Reviewer xnHx

1. **The analysis about using simulated annealing as an optimizer**.
   * Section 4.1.7  Parameter Sensitivity Experiments (Line515 - 517)
   * Appendix A.3.6 More Parameter Sensitivity Analysis (Line1149 - 1155)
   * Table 4 (Line 1188-1209)
2. **Clarify the relationship between MASS and traditional factor-mining approaches.**
   * Section 2.1: Investment analysis: (Line 100)
   * Section 4 Baselines  (Line 322)
   * Section 4.1.3: Update on backtesting curves  Figure 2 (Line 378-396)
   * Update on Table 1 (Line 287-288 Line 298-299)
   * Update on Table 7 (Line 1252-1253, Line 1267-1268)
   * Table 8 (Line 1306-1307, Line1319-1320)



## Reviewer ym2m

1. **The testing performance of the baseline methods at the beginning of 2025.**
   * Section 4.1.2  Experiments on Data Leakage concern (Line 373)
   * Update on Table 1 (Line 297-302)
   * Update on Table 7 (Line 1265-1273)
   * Table 8 (Line 1317-1324)
2. **Analysis or observation from the trading strategy perspective.**
   * Section 4.1.6: stability and agent distribution experiment (Line 481-485)
   * Appendix 3.5: Case study (Line 1074-1078, Line 1118-1130)
   * Figure 6: (Line 1080-1115)

Once again, we thank you for your time and valuable feedback. We hope our revisions have successfully addressed your concerns.

Sincerely

Authors

---

### Author Response · Authors · 2025-11-28
**Timeline of Review Process**

## **Timeline of Review Process (AOE Time)**

### **Phase 1: Author Rebuttal (First Response)**

*   **17 Nov 2025, 04:19:00** - Authors post their first response to **all** Reviewers, followed by detailed point-by-point replies.
*   **17 Nov 2025, 04:55:00** - Authors post a general comment summarizing the strengths of the paper and outlining their response strategy.
*   **24 Nov 2025, 20:37:00** - Authors post a "Revision Summary" detailing all major changes made to the manuscript in response to all reviewers.

### **Phase 2: Reviewer-Author Interaction and Further Clarifications**

*   **18 Nov 2025, 12:25:00** - Reviewer **ym2m** posts a response to the authors, confirming their concerns have been adequately addressed and stating they have updated their rating. *(Note: raised from 4 to 8 on November 18th, 12:26 AOE).* [Evidence](https://openreview.net/revisions?id=xYxLe8ft6Z)
*   **18 Nov 2025, 14:06:00** - Authors post a reply to thank Reviewer **ym2m** for the positive feedback and score update.

### **Phase 3: Author Follow-ups**

*   **25 Nov 2025, 18:46:00** - Authors post a follow-up comment gently requesting further feedback from Reviewer **QTnN, EMCk, xnHx**.
*   **27 Nov 2025, 17:39:00** - Authors post a second follow-up to Reviewer **QTnN, EMCk, xnHx**, summarizing the changes made and asking if their concerns were addressed.

---

### Author Response · Authors · 2025-12-03
**Clarification of our rebuttal and score**

## Dear AC, SAC, PC, Reviewers,

Much before this OpenReview bug occurred (02:00 AOE on November 27), we resolved the issues raised by reviewer **ym2m** through rebuttal:

* **ym2m**'s score was ultimately raised from 4 to 8 (12:26 AOE on November 18) [\[Evidence from Official History\]](https://openreview.net/revisions?id=xYxLe8ft6Z).

These score increases were based on our multi-turn conversation and resolution of their issues, which can be verified through the rebuttal history.

As of the OpenReview bug occurrence, we have not yet received any replies from the other reviewers. However, we try our best to address every reviewer's points one by one:

* For Reviewer **QTnN**(with score 6) and Reviewer **EMCk** (with score 4), we addressed their concerns about generalizability and interpretability by providing extensive new experiments, including cross-market validation on US stocks, new SOTA baselines, sensitivity analysis on optimizer, LLM backbone and prompt design, more investigation on scaling effect,  and a detailed case study.
* For Reviewer **xnHx** (with score 4), we clarified several core misunderstandings regarding our optimization method and the market disagreement hypothesis, and we added the requested factor-model baselines to our experiments, which further strengthened our results.

Following our rebuttal, our paper’s score was updated to **[8, 6, 4, 4] on 12:26 AOE on November 18, 9 days before the security breach**. As of 02:00 AOE on November 27, our paper was ranked in the top **18%**.

We wish to ensure that this situation is accurately reflected, and we respectfully request that you take this information into consideration when making your final decision.

Sincerely,

The Authors

---

### Meta-Review · Area_Chair_6TKT · 2026-01-07

**Summary:**

1. No Theoretical Justification for Scaling Effect: The central claim of performance scaling with agent count remains an empirical observation only. Post-rebuttal additions provide more data but no formal theory (e.g., on agent diversity or error cancellation), limiting insight into emergent behaviors.
2. Inadequate Discussion of Real-World Risks: In high-stakes finance, LLM agents pose dangers like hallucinations or failures in extreme markets. The paper mentions costs and open-sourcing but lacks meaningful analysis of these practical/deployment risks, regulatory issues, or safety mitigations.

These issues persist post-rebuttal and, combined with, the borderline reviewer scores. The empirical results are commendable, but the paper falls short of providing deep insights or responsible deployment guidance expected at ICLR.

**Reviewer Concerns:**

Addressed Concerns:

1. Generalizability across markets and regimes: Authors added comprehensive experiments on US markets (Nasdaq 100, S&P 500) using public datasets, covering 2023 and Q1 2025, demonstrating cross-market robustness.

2. Missing strong baselines, especially factor models: Added FactorVAE and HireVAE as baselines across all settings (Chinese and US markets, multiple periods). MASS consistently outperforms them.

3. Data leakage / robustness on unseen 2025 data: Extended all baselines to Q1 2025 data in both markets; updated tables show sustained outperformance, effectively ruling out leakage.

4. Over-reliance on prompt engineering and limited novelty in aggregation: Demonstrated prompts are programmatically generated (not hand-crafted); added experiment with radically rephrased prompts showing unchanged performance; emphasized dynamic backward optimization as the novel component.

Outstanding concerns:

1. Theoretical justification for scaling effect: the authors provide more empirical insight but no formal theory (e.g., diversity or error cancellation models).

2. Real-world deployment complexity and practical risks: Authors highlighted open-sourcing and low costs but did not deeply discuss broader risks (e.g., LLM hallucinations in high-stakes trading, regulatory compliance, or failure modes in extreme market events).

**Reviewer Scores:**

The reviewer xnHx and EMCk may maintain their negative scores.

---

### Decision · Program_Chairs · 2026-01-26

Reject